# A plant cytorhabdovirus modulates locomotor activity of insect vectors to enhance virus transmission

Dong-Min Gao[1], Ji-Hui Qiao[1], Qiang Gao[1,2], Jiawen Zhang[3], Ying Zang[1], Liang Xie[1], Yan Zhang[3], Ying Wang [2], Jingyan Fu[3], Hua Zhang [3], Chenggui Han [2] & Xian-Bing Wang [1]

Transmission of many plant viruses relies on phloem-feeding insect vectors. However, how plant viruses directly modulate insect behavior is largely unknown. *Barley yellow striate mosaic virus* (BYSMV) is transmitted by the small brown planthopper (SBPH, *Laodelphax striatellus*). Here, we show that BYSMV infects the central nervous system (CNS) of SBPHs, induces insect hyperactivity, and prolongs phloem feeding duration. The BYSMV accessory protein P6 interacts with the COP9 signalosome subunit 5 (LsCSN5) of SBPHs and suppresses LsCSN5-regulated de-neddylation from the Cullin 1 (CUL1), hereby inhibiting CUL1-based E3 ligases-mediated degradation of the circadian clock protein Timeless (TIM). Thus, virus infection or knockdown of *LsCSN5* compromises TIM oscillation and induces high insect locomotor activity for transmission. Additionally, expression of BYSMV P6 in the CNS of transgenic *Drosophila melanogaster* disturbs circadian rhythm and induces high loco-motor activity. Together, our results suggest the molecular mechanisms whereby BYSMV modulates locomotor activity of insect vectors for transmission.

Insects are vectors of many parasites causing various diseases in animals and plants. Increasing evidence has demonstrated that these parasites have evolved diverse effectors to modulate insect behaviors for their horizontal spread in nature[1]. Transmission of insect-borne plant pathogens including viruses, fungi, bacteria, and phytoplasmas heavily relies on their insect vectors[1]. It is well known that plant viruses directly or indirectly modulate behavior of their insect vectors for efficient transmission[2,3]. During indirect mod-ulation, plant viruses render infected plants to become more attractive to insect vectors by changing volatile odor compounds, nutrition, visual appearance, or defense profiles[2,4]. Although it is a widespread phenomenon, we are just beginning to understand the underlying mechanisms how plant viruses modify plant

physiological states to attract insect vectors indirectly[3]. For instance, the C2 protein of tomato yellow leaf curl virus (TYLCV) and the NSs protein of tomato spotted wilt tospovirus (TSWV) inhibit MYC-regulated jasmonic acid (JA) signaling to enhance vec-tor attraction[5,6]. The cucumber mosaic virus 2b protein directly interacts with JAZ proteins to inhibit JA-mediated herbivore defense of host plants[7]. The begomovirus βC1 protein modulates plant immunity, which is beneficial for infection of begomovirus and its whitefly vectors but deters non-vector insects[8]. These findings represent indirect effects of virus components on insect behavior manipulation through changing plant phenotypes and inducing insects to preferentially settle on infected plants for virus acquisition[3,9]. However, it is a far less described and functional

[1]State Key Laboratory of Plant Environmental Resilience, College of Biological Sciences, China Agricultural University, Beijing 100193, China. [2]Ministry of Agriculture and Rural Affairs Key Laboratory of Pest Monitoring and Green Management, College of Plant Protection, China Agricultural University, Beijing 100193, China. [3]State Key Laboratory of Animal Biotech Breeding, College of Biological Sciences, China Agricultural University, Beijing 100193, China. ✉e-mail: wangxianbing@cau.edu.cn

explanation for how persistently transmitted plant viruses directly regulate insect behaviors when they propagate within the vectors.

Some parasites can spread into the central and peripheral nervous systems of insect vectors to regulate insect behaviors and performances, termed neuroparasitology[10]. For instance, the uridine 5′-diphosphate (UDP)–glucosyltransferase of baculoviruses render infected gypsy moths to climb to tree foliages, facilitating virions release from insect cadavers and are acquired by new insect vectors[11,12]. Besides, Kakugo RNA, a novel picorna-like virus, was only detected in the brains of aggressive honey bees[13], implying that honeybee aggressive behaviors probably enhance virus horizontal transmission. Most insect-borne plant viruses are transmitted along with salivary secretion during insect feeding on host plants. Therefore, longer feeding time and more food intake can enhance virus transmission probability. For instance, TSWV drives thrips to take more food than healthy thrips, which facilitate virus transmission along feeding[14]. Similarly, viruliferous whiteflies with TYLCV have more food intake than healthy whiteflies for virus transmission[15]. Besides, TYLCV can cause apoptotic neurodegeneration in whitefly, and drive the movement of viruliferous whiteflies from infected plants to surrounding healthy plants for virus spread[16]. Rice gall dwarf virus (RGDV), a devastating rice reovirus, inhibits the secretion of a calcium-binding protein, leading to more saliva secretion into host plants for virus transmission[17]. However, to date, the mechanisms of directly modulation of insect feeding behavior and the responsible virus components have not yet been studied in previous studies.

Most insects exhibit high locomotor activity before sunrise and sunset (termed as morning and evening peaks), which is regulated by endogenous circadian clocks[18]. The molecular machinery of circadian clock has been extensively studied in *Drosophila melanogaster*[19]. Two transcription factors, CLOCK (CLK) and CYCLE (CYC), form a heterodimer and promote transcription of *period* (*per*) and *timeless* (*tim*) genes in late day to early night[20–22]. Subsequently, the PER and TIM proteins gradually accumulate during the night, form heterodimers late night, and move into the nucleus to repress transcription of CLK-CYC heterodimer[20,21]. Light can entrain the clock system through degrading TIM by light-activated CRY and JETLAG (JET)[23–26]. The F-box protein, JET, mediates light-induced TIM degradation through triggering ubiquitination of TIM, which requires the COP9 signalosome (CSN) to remove Nedd8 moiety from the scaffold protein cullin1 of the SCF$^{JET}$ complex[27]. The *per/tim* oscillatory loop triggers a rhythmic expression of various clock-related genes for modulation of insect behavior and physiological functions[28]. However, whether and how insect-borne plant viruses modify circadian rhythms of insect vectors remains elusive.

The family Rhabdoviridae consists of 20 genera with ecologically diverse members infecting mammals, plants, insects, fishes, birds, and reptiles[29]. Two model rhabdoviruses, *Vesicular stomatitis virus* (VSV) and *Rabies virus* (RABV), can invade the central nervous system (CNS) and induce hyperactivity in infected animals[30–33]. Most plant rhabdoviruses are transmitted by insect vectors in a circulative-propagative manner. Several plant rhabdoviruses, including rice yellow stunt virus (RYSV), rice stripe mosaic virus (RSMV), maize mosaic virus (MMV), and maize fine stripe virus (MFSV), have been shown to spread into the CNS of insect vectors[34–37]. Recently, reverse genetic systems been developed in several plant rhabdoviruses, including sonchus yellow net virus (SYNV)[38], barley yellow striate mosaic virus (BYSMV)[39], northern cereal mosaic virus (NCMV)[40], and MMV[41]. These systems provide genetic approaches for investigating virus–insect–plant interactions[42,43].

*Barley yellow striate mosaic virus* (BYSMV), an important member of the *Cytorhabdovirus* genus, infects >25 species of monocots worldwide. BYSMV is persistently transmitted by the small brown planthoppers (SBPHs, *Laodelphax striatellus*). The BYSMV genome encodes five accessory proteins (P3, P4, P5, P6, and P9) interspersing

into five conserved structural proteins (N, P, M, G, and L)[44]. We have established reverse genetics systems of BYSMV and applied these systems to studies of the tripartite interactions of virus–insect–plant[43,45–48]. Recently, we found that BYSMV P6 inhibits JA signaling in host plants and indirectly affects insect attractiveness[49]. Here, we undertook experiments to identify a new function of BYSMV P6 in regulating locomotor activity of their insect vectors. We here show that BYSMV P6 interferes with degradation of TIM and directly modulates locomotor activity of insect vectors for virus transmission.

## Results

### BYSMV infects the central nervous system (CNS) and induces high locomotor activity in the small brown planthopper (SBPH, *Laodelpha striatellus*)

BYSMV is an insect-borne cytorhabdovirus and multiplies within its insect vectors, SBPHs. We have previously revealed the presence of BYSMV in alimentary canals and salivary glands of viruliferous SBPHs[50,51]. Here, we determined whether BYSMV spread into the CNS of SBPHs. To this end, we used a recombinant BYSMV expressing red fluorescent protein (BY-RFP), in which the *RFP* gene with the gene junction sequences was inserted between the N and P genes of BYSMV (Fig. 1a)[39]. BY-RFP can express RFP during virus infection, allowing us to visualize authentic infection of BYSMV in living tissues of SBPHs. At 12 days (d) after a 2-d-inoculation period on BY-RFP-infected barley plants[52], intense RFP fluorescence was observed in ~40% of inoculated SBPHs, but not in non-inoculated controls (Supplementary Fig. 1a). Immunoblotting analyses confirmed accumulation of RFP and BYSMV N in BY-RFP-infected SBPHs (Supplementary Fig. 1b).

To visualize the presence of BY-RFP in the nervous systems, we performed histological analyses on healthy SBPHs and BY-RFP-infected SBPHs at 12 dpi. The results showed that RFP fluorescence was present in the CNS of viruliferous SBPHs, but not in those of healthy SBPHs (Fig. 1b). RFP fluorescence was observed in all the dissected CNS of viruliferous SBPHs, rather than in those controls (Fig. 1c, Supplementary Fig. 2). In addition, these dissected brain tissues were stained with the specific rabbit antibodies against the N protein and then incubated with alex-488-coupled rabbit secondary antibody (GFP) for observation under a fluorescence microscopy. As expected, RFP fluorescence from BY-RFP was overlapped with the viral N protein in the viruliferous SBPHs, rather than in controls (Supplementary Fig. 3), indicating that RFP fluorescence can used as an indicator of virus infection sites. Immunoblotting analysis showed that RFP and BYSMV N accumulated in brain tissues of BY-RFP-infected SBPHs (Fig. 1d). These results indicate that BY-RFP can infect the CNS of SBPHs as other plant rhabdoviruses[34–37].

Animal rhabdoviruses commonly cause aggressive behavior in their animal hosts[32,33]. To address the hypothesis that BY-RFP infection increases activity in SBPHs, we used the Drosophila Activity Monitor 2 (DAM2) system to monitor the locomotor activity of non-inoculated or viruliferous SBPHs under light–dark (LD) cycles. Notably, the locomotor activity of BY-RFP-infected SBPHs was higher than that of non-inoculated controls (Fig. 1e, f). In constant darkness (DD), viruliferous SBPHs still exhibited higher locomotor activity than non-inoculated SBPHs (Supplementary Fig. 4a, b). Whereas, there was no obvious difference between the two kinds of SBPHs in constant light (LL) treatment (Supplementary Fig. 4c, d), indicating constant light treatment abolished the effect of virus infection on insect locomotor activity.

Collectively, these results demonstrate that BY-RFP infects the CNS and increases locomotor activity in SBPHs.

### BYSMV infection increases feeding duration and food intake amounts of SBPHs

Feeding behavior of insect vectors plays crucial roles in transmission of plant viruses. Therefore, we examined whether the BYSMV-induced

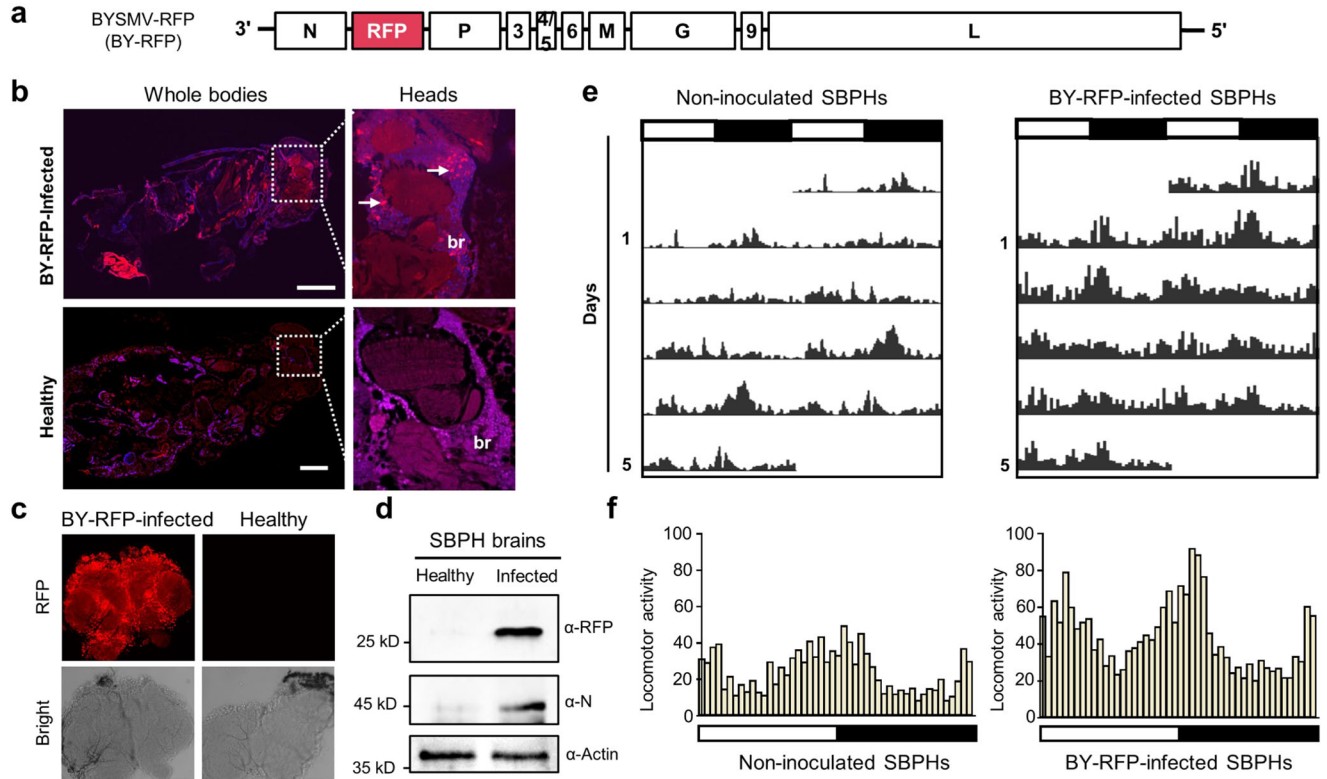

**Fig. 1 | Barley yellow striate mosaic virus (BYSMV) infects the central nervous system (CNS) and increases locomotor activity in *Laodelphax striatellus*. a** The schematic diagram of the recombinant BYSMV-RFP (BY-RFP) virus for indicating virus real-time infections in insect vectors. The *RFP* gene with the gene junction sequence was inserted between the *N* and *P* genes of BYSMV, and the resultant BY-RFP can express RFP protein during virus infection in living tissues of SBPHs.
**b** Histological analysis and confocal image stacks showing the presence of BY-RFP in the CNS of viruliferous SBPHs infected by BY-RFP for 12 d. The arrows indicated BY-RFP-infected foci in neuropils. The tissues were stained with DAPI to indicate nuclei. Scale bars, 200 μm. br, brain. **c** Representative confocal image showing RFP fluorescence in the dissected brain of BY-RFP-infected SBPHs at 15 dpi. Scale bars,

100 μm. **d** Immunoblotting analyzing accumulation of RFP and BYSMV N proteins in dissected brains from healthy or BY-RFP-infected SBPHs. **e** The locomotor activity of non-inoculated and BY-RFP-infected SBPHs in a light/dark (LD) cycle. The column numbers represent total across times of SBPHs ($n = 32$) in the middle of tubes monitored by infrared beam in 5 d. White and black bars indicate light and dark phases, respectively. **f** Comparison of the circadian locomotor activity of non-inoculated or BY-RFP-infected SBPHs in LD cycles. Histograms represent the distribution of activity counts (*y* axis) every 30 min of indicated SBPHs ($n = 32$) through 24 h, averaged over 5 LD days. **b**–**d** were repeated three times independently with similar results. Source data are provided as the Source Data file.

high locomotor activity affected feeding behavior of SBPHs. Persistently transmitted plant viruses require efficient feeding on phloem tissues for virus acquisition and inoculation. The electrical penetration graph (EPG) technique is a powerful tool for recording feeding behaviors of sap-sucking insects on host plants[53]. We used the EPG assay to measure the feeding durations of healthy and viruliferous SBPHs on healthy barley plants. Generally, the EPG waveforms can be divided into several kinds of periods: non-penetration (Np), penetration (N1, N2, and N3), phloem-feeding (N4a and N4b), and xylem feeding (N5) (Fig. 2a). The EPG waves of healthy SBPHs showed long periods of non-penetration (NP) and penetration (N1, N2, and N3), but relative short feeding time in phloem (N4a and N4b) (Fig. 2b, Supplementary Fig. 5). By contrast, BY-RFP-infected SBPHs spent more feeding time in phloem (N4a and N4b) (Fig. 2c, Supplementary Fig. 5). We calculated feeding duration in phloem from 8-h inoculation time with healthy barley plants, showing that BY-RFP infection induced an extended phloem feeding time by approximately 2-fold compared with mock-treated SBPHs (Fig. 2d).

Honeydew secretion has been used to indicate food intake amounts of sap-sucking insects[54]. To test whether virus infection influenced food intake of SBPHs, we incubated healthy or BY-RFP-infected SBPHs within an inverted transparent plastic cup over a filter paper (Fig. 2e). After 2 d, the filter papers were stained with ninhydrin, and the areas were used to indicate protein amounts of honeydew (Fig. 2e). The statistical analysis indicates that BY-RFP-infection

significantly increases honeydew excretion of SBPHs compared with mock-treated SBPHs (Fig. 2e, f). These results suggest that BY-RFP infection improves the food intake amount of viruliferous SBPHs.

Collectively, these results clearly demonstrate that BY-RFP infection increases feeding duration and food intake amount of SBPHs from phloem tissues.

## BYSMV infection disrupts the TIM oscillation and circadian rhythms by inhibiting TIM degradation

The results above have demonstrated that BY-RFP infection modulates the locomotor activity and feeding behavior of SBPHs. We reasoned that the behavior alteration may be attributed to an abnormal circadian rhythm induced by BY-RFP infection. To test this hypothesis, we examined oscillations of the clock genes, *period* (*per*) and *timeless* (*tim*), through RT-qPCR to measure their relative transcripts at five time points (Zeitgeber time, ZT 0, 6, 12, 18, and 24) under LD cycles. Notably, oscillation of *tim* and *per* transcripts was significantly inhibited in viruliferous SBPHs compared to robust oscillation in mock-treated SBPHs (Fig. 3a, b). In addition, mRNA oscillations of *clock* (*clk*) and *cycle* (*cyc*), another two core transcription factors, were also attenuated in BY-RFP-infected SBPHs but maintained in mock-treated SBPHs (Fig. 3c, d).

Immunoblotting analyses revealed that the TIM protein accumulation of mock-treated SBPHs decreased significantly after light treatment in the early morning (Fig. 3e, left panel). By contrast, the TIM

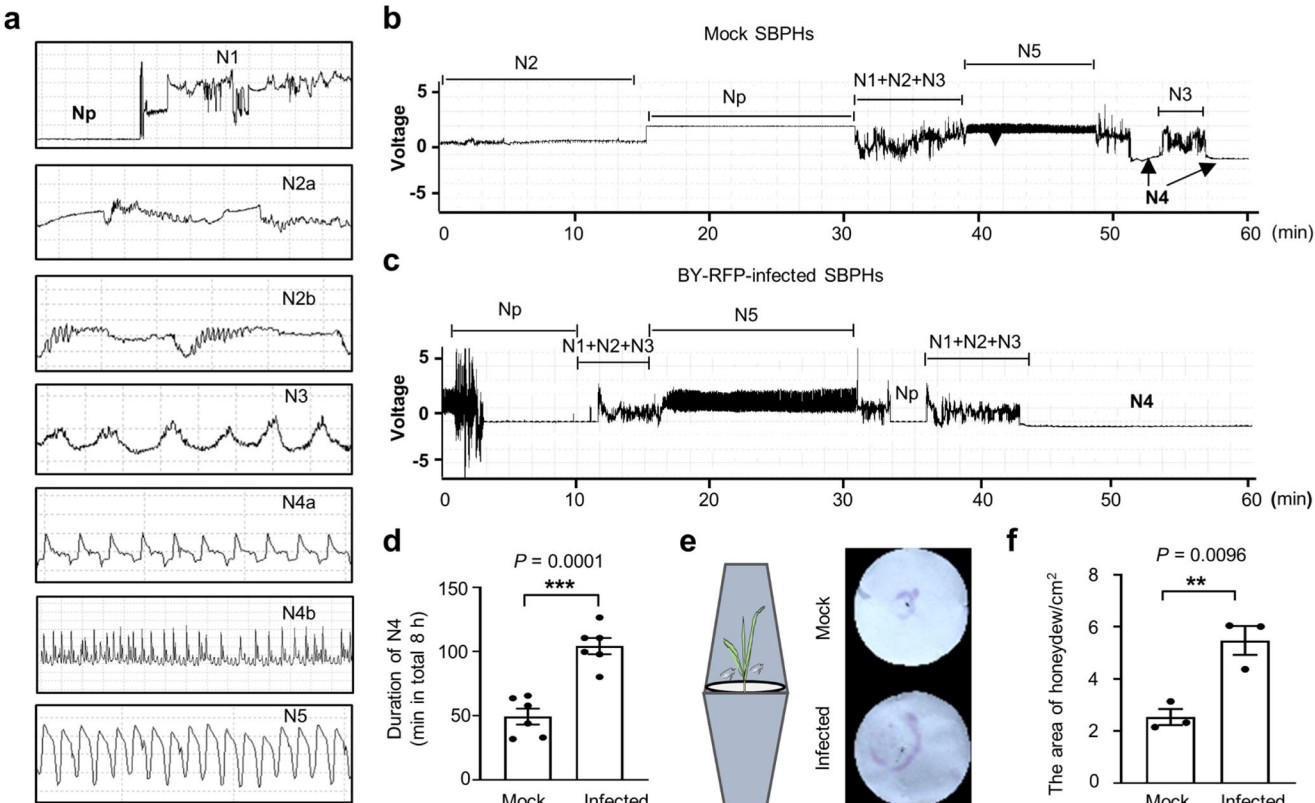

**Fig. 2 | BYSMV infection increases feeding duration and food intake amounts of SBPHs. a** Typical electrical penetration graph (EPG) signals of insect feeding on barley plants, including non-penetration (Np), penetration (N1, N2, N3), phloem feeding (N4a and N4b), and xylem feeding (N5). **b** The representative EPG waveforms of mock buffer-treated SBPHs feeding on barley plants. **c** The representative EPG waveforms of viruliferous SBPHs feeding on barley plants. **d** Quantitative assessment of the feeding duration time of mock or viruliferous SBPHs in total 8 h. Error bars represent SEM ($n = 6$ biologically independent SBPHs). ***$P < 0.001$

(two-sided $t$ test). **e** Diagrammatic illustration of the honeydew bioassay. Mock or BY-RFP-infected SBPHs (10 insects each) were incubated within an inverted transparent plastic cup over a filter paper (9 cm in diameter) around a healthy barley plant. After 2 d, the filter papers were stained with ninhydrin. **f** Mean areas of honeydew excretion by mock or BY-RFP-infected SBPHs. The areas of stained honeydew were calculated by the Image J software. Error bars represent SEM ($n = 3$ biologically independent experiments, 10 SBPHs per experiment). **$P < 0.01$ (two-sided $t$ test). Source data are provided as the Source Data file.

protein maintained at similar levels through the whole LD cycles in BY-RFP-infected SBPHs (Fig. 3e, right panel), implying that BY-RFP infection disturbed the oscillation of the TIM protein under LD cycles.

In *Drosophila*, light-induced rapid TIM degradation is a key step of molecular clock entrainment in LD cycles[55]. Therefore, we examined TIM degradation in mock or viruliferous SBPHs for 0- and 10- light treatment transferred from the darkness at ZT15. Immunoblotting analysis results consistently confirmed that light treatment induced obviously decreased accumulation of TIM in mock-treated SBPHs, but not in BY-RFP-infected SBPHs (Fig. 3f). Collectively, these results indicate that BY-RFP infection disrupts TIM degradation, oscillation, and circadian rhythms of insect vectors.

## BYSMV P6 interacts with the LsCSN5 subunit and interfered with de-neddylation of CUL1 by LsCSN5

Previous studies have demonstrated that the CSN5 component of the COP9 signalosome is essential for light-mediated TIM degradation and clock resetting in *Drosophilia*[27]. We recently found that BYSMV P6 interacts with the CSN5 (HvCSN5) of barley to subvert plant JA signaling[49]. Moreover, the 16th Ile (Ile16) residue of BYSMV P6 is required for the specific interaction with HvCSN5[49]. Since COP9 and CSN5 are conserved in all eukaryotes[56,57], we assumed that BYSMV P6 may interact with the CSN5 protein (LsCSN5) of SBPHs, which shares 60.0% identity with HvCSN5 (Supplementary Fig. 6a, 6). To test this hypothesis, we performed yeast two-hybrid (Y2H) assays to confirm the P6–LsCSN5 interaction. LsCSN5 was fused to the C terminus of the GAL4 DNA binding domain (BK), while BYSMV P6 and P6$^{I16A}$ were fused

to the C terminus of the GAL4 activation domain (AD). All yeasts expressed protein combinations grew on the SD/-Trp-Leu medium. Only yeasts expressing AD-P6 and BK-LsCSN5, rather than other protein combinations, were able to proliferate on SD/-Trp-Leu-His-Ade selection medium with 5 mM 3-amino-1,2,4-triazole (3-AT) (Fig. 4a), indicating that P6, rather than P6$^{I16A}$, interacts with LsCSN5 in yeast.

We further confirmed the interaction of LsCSN5 with the P6 protein expressed from BYSMV-infected SBPHs. The purified GFP-His or LsCSN5-His from *E. coli* were incubated with total protein extracts from BY-RFP-infected SPBHs and precipitated with anti-His agarose beads for immunoblotting analysis with anti-P6 and anti-His antibodies. The results show that only LsCSN5-His, rather than GFP-His, can be co-precipitated with the BYSMV P6 protein expressed from BY-RFP infections (Fig. 4b). We further performed GST pull-down assays to verify the in vitro P6–LsCSN5 interaction. LsCSN5-His was incubated with GST, GST-P6, or GST-P6$^{I16A}$, and precipitated with anti-GST beads. Immunoblotting analysis showed that LsCSN5-His was only co-precipitated with GST-P6, rather than GST or GST-P6$^{I16A}$ (Fig. 4c).

The COP9 CSN5 component is responsible for removing the Nedd8 moiety from the scaffold protein CUL1 of the Skp1/Cullin1/F-box (SCF) E3 ligase complexes[58]. Thus, we next tested whether the P6–LsCSN5 interaction interfered with de-neddylation of CUL1. Firstly, we examined accumulation of LsCUL1 in healthy or BY-RFP-infected SBPHs, and showed that BY-RFP infection led to significantly increased ratio of CUL1$^{Nedd}$ (CUL1 with Nedd8 moiety) to CUL1 (Fig. 4d), indicating that BY-RFP interfered with de-neddylation of CUL1.

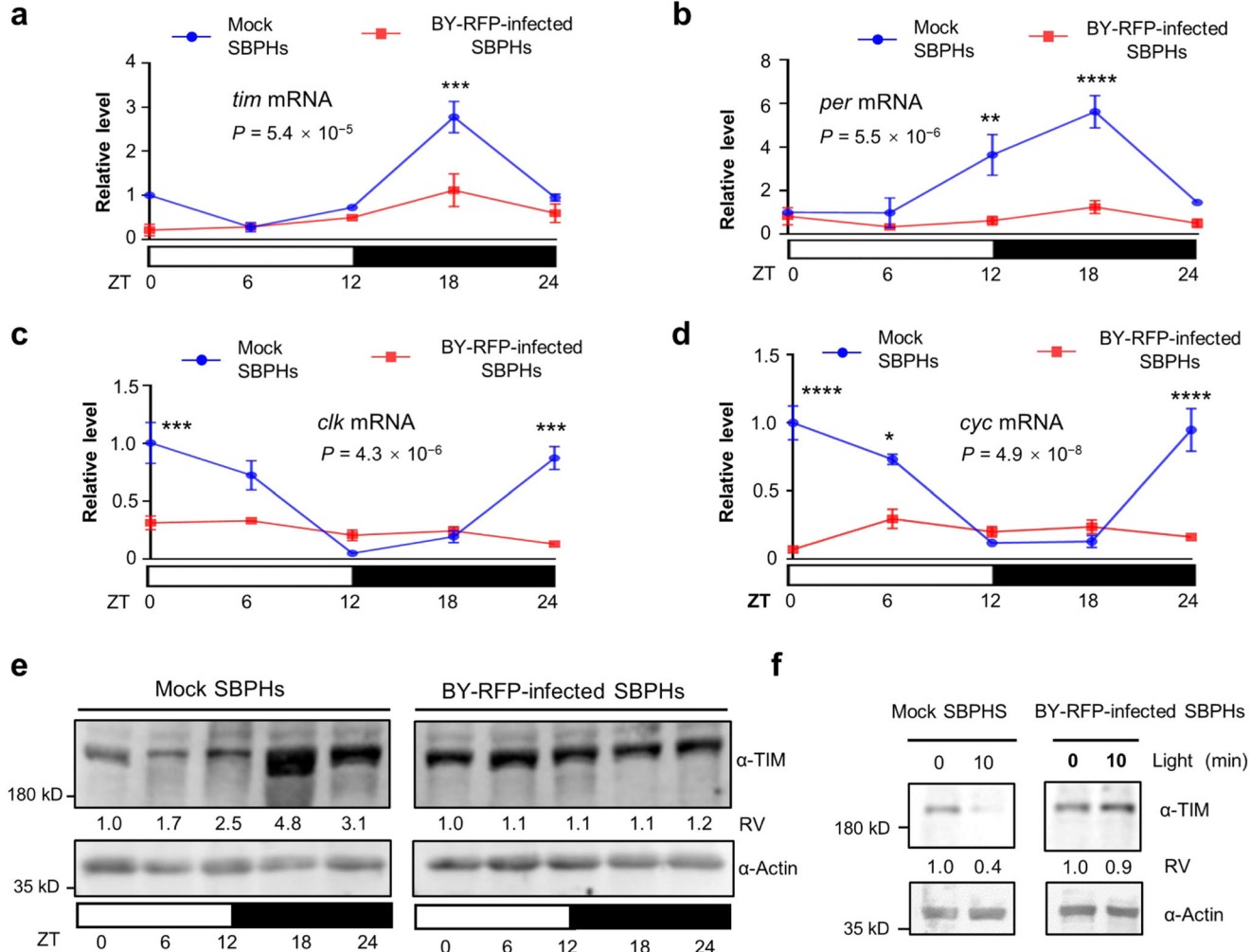

**Fig. 3 | BYSMV infection disrupts the TIM oscillation and circadian rhythms by inhibiting light-dependent TIM degradation. a**–**d** RT-qPCR analyzing the relative levels of *tim* (**a**), *per* (**b**), *clk* (**c**), and *cyc* (**d**) mRNAs in the mock-treated or BY-RFP-infected SBPHs under LD cycles. The results are nomalized to the accumulation of mRNA at ZT0 of mock-treated SBPHs that was set as 1. The SBPH *actin1* gene served as an endogenous control. The error bars indicate SEM (*n* = 3 biologically independent experiments, 20 SBPHs per experiment). Two-way ANOVA followed by Tukey's test was performed to investigate the main effects of BY-RFP infection on mRNA levels. Differences were considered significant at *P* < 0.05. Asterisks indicate significant differences between mock-treated and BY-RFP-infected SBPHs at the same indicated time points. *\*P* < 0.05, *\*\*P* < 0.01, *\*\*\*P* < 0.001, and *\*\*\*\*P* < 0.0001. **e** Immunoblotting analyzing accumulation of the TIM protein in the mock or BY-RFP-infected SBPHs under LD cycles. **f** Immunoblotting analyzing accumulation of the TIM protein in dissected heads of mock- or BY-RFP-infected SBPHs treated with 0- and 10-min light after ZT15. **e**, **f** the TIM protein was detected with anti-TIM antibodies. Actin was detected with anti-Actin antibodies as a loading control. Relative values (RV) of TIM were calculated from band densities and normalize to against Actin accumulation (TIM/Actin). The values in the ZT24 acted as 1. These experiments were repeated three times independently with similar results. Source data are provided as the Source Data file.

To examine whether LsCSN5 was required for de-neddylation of CUL1, *LsCSN5* dsRNA (*dscsn5*) were synthesized in vitro and injected into the thorax of SBPHs to interfere with endogenous *LsCSN5*, and the gfp dsRNA (*dsgfp*) served as a negative control. As expected, SBPHs injected with *dscsn5* accumulated relative higher ratio of CUL^Nedd to CUL1 compared with the *dsgfp* control (Fig. 4e). Collectively, these results confirmed the P6–LsCSN5 interaction, which negatively affected LsCSN5-mediated de-neddylation of endogenous CUL1.

**Knockdown of *LsCSN5* inhibits light-dependent TIM oscillation and improves virus transmission capacity in SBPHs**
The results above have demonstrated that BY-RFP infection and knockdown of *LsCSN5* inhibited de-neddylation process of CUL1, which would cause defective SCF E3 ligase complexes. Moreover, CUL1-based E3 ligases can mediate degradation of the circadian clock protein TIM. Therefore, we examined whether LsCSN5 was involved in TIM degradation and circadian rhythm of SBPHs. At 2 d

after injection, the *LsCSN5* transcript was significantly down-regulated in *dscsn5*-injected SBPHs compared with those of *dsgfp* (Supplementary Fig. 7a). We further examined whether knockdown of *LsCSN5* modulated the circadian rhythm of SBPHs in LD cycles. Oscillation of the TIM protein was maintained in *dsgfp*-injected SBPHs under LD cycles (Fig. 5a, left panel). By contrast, accumulation of the TIM protein in *dscsn5*-injected SBPHs was similar in all the stages of LD cycles, indicating that knockdown of *LsCSN5* inhibited oscillation of the TIM protein (Fig. 5a, right panel). Moreover, oscillation of the *per* and *tim* transcripts were significantly disturbed in *dscsn5*-injected SBPHs compared to those of *dsgfp* (Fig. 5b). We further treated *dsgfp*- or *dscsn5*-injected SBPHs with mock or 10-min light for immunoblotting analyses with anti-TIM or anti-Actin antibodies. The results showed that *dscsn5* injection suppressed light-dependent TIM degradation, but *dsgfp* injection did not (Fig. 5c). These results indicate that the knockdown of *LsCSN5* disturbs TIM degradation, oscillation, and circadian rhythms.

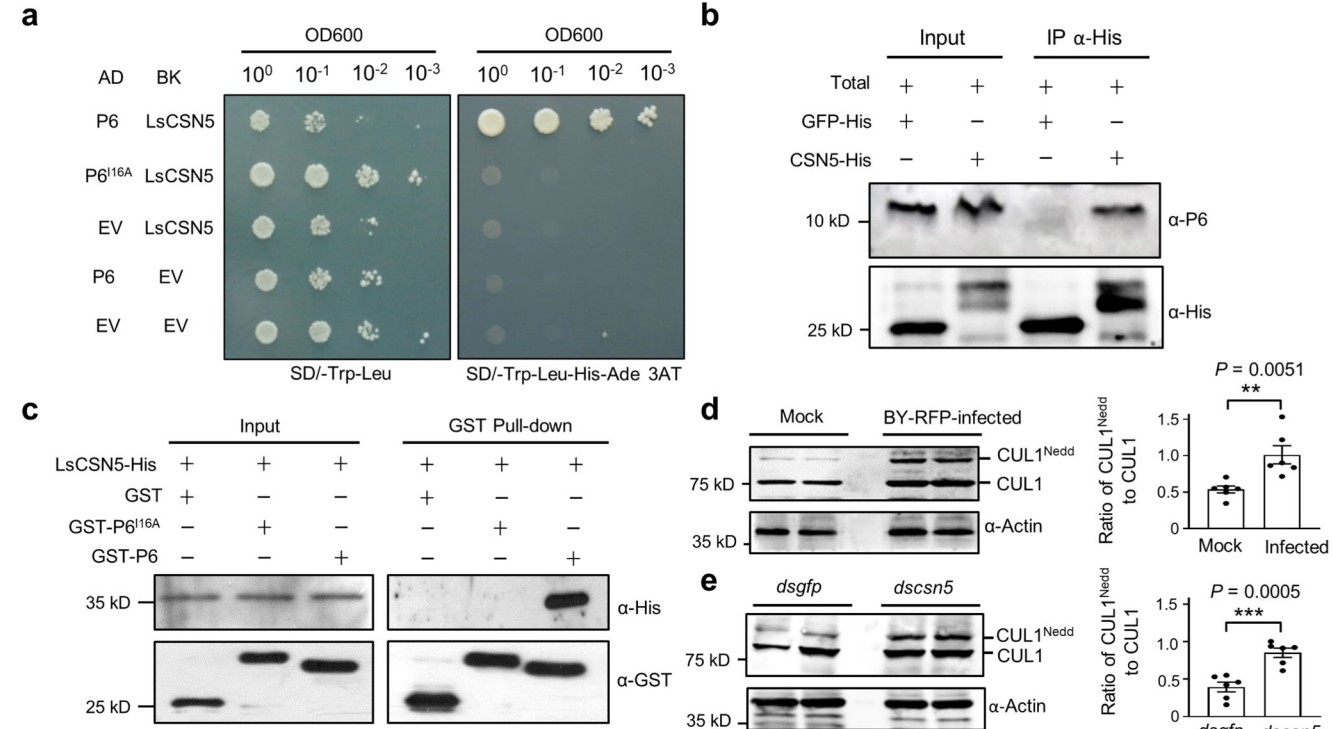

**Fig. 4 | BYSMV P6 interacts with the LsCSN5 subunit and interfered with the LsCSN5-mediated de-neddylation of CUL1. a** Yeast two-hybrid assays for determining protein interactions between LsCSN5 with P6 or P6^II6A. The empty pGADT7 and pGBKT7 vectors served as negative controls. AD, GAL4 activation domain; BK, GAL4 DNA binding domain, 3-AT (5 mM), 3-amino-1,2,4-triazole. **b** The LsCSN5-His protein pulled down the P6 protein from total protein samples of viruliferous SBPHs. Immunoblotting was performed using anti-His and anti-P6 antibodies. GFP-His was used as negative control. **c** GST pull-down assays showing LsCSN5-P6 interactions in vitro. LsCSN5-His was incubated with GST-P6, GST-P6^II6A or GST and immunoprecipitated with Glutathione-Sepharose beads. The pull-down and input proteins were detected by immunoblotting with anti-GST (α-GST) and anti-His (α-His) antibodies. **d** Immunoblotting detecting accumulation of LsCUL1 in mock-treated or viruliferous SBPHs. **e** Immunoblotting detecting accumulation of LsCUL1 in SBPHs injected with *dsCSN5* or *dsgfp*. **d**, **e** positions of CUL1^Nedd and CUL1 were indicated. The ratios of CUL1^Nedd to CUL1 were calculated and showed on right panels. Error bars indicates SEM of six independent repeats (20 insects each repeat). **$P < 0.01$; ***$P < 0.001$ (two-sided $t$ test). All experiments were repeated three times independently with similar results. Source data are provided as the Source Data file.

We next tested whether knockdown of *LsCSN5* affected locomotor activity of SBPHs using the DAM2 assay. As expected, the locomotor activity of *dscsn5*-injected SBPHs was higher than those of *dsgfp*-injected SBPHs (Supplementary Fig. 7b, c). The calculated average activity counts per 30 min of *dscsn5*-injected SBPHs were significantly higher than those of *dsgfp*-injected SBPHs in the ZT 0–6, 6–12, 18–24 of LD cycles (Fig. 5d). Using the honeydew detection assays as shown in Fig. 2e, *dscsn5*-injected SBPHs secreted significantly more amount of honeydew than *dsgfp*-injected SBPHs (Fig. 5e), indicating enhanced food intake amount of *dscsn5*-injected SBPHs.

BY-RFP-infected SBPHs were further injected with *dscsn5* or *dsgfp*, and both SBPHs accumulated similar levels of RFP and BYSMV N proteins at 6 dpi (Fig. 5f), as well as similar intensity of RFP fluorescence (Fig. 5g), indicating that knockdown of *LsCSN5* did not interfere with BY-RFP infection in SBPHs. Then, we incubated these SBPHs with healthy barley plants for 1-day inoculation period. At 12 dpi, the barley plants inoculated with *dscsn5*-injected SBPHs exhibited higher intensity of RFP fluorescence than those inoculated with *dsgfp*-injected SBPHs (Fig. 5h). Time course also showed that feeding of *dscsn5*-injected SBPHs improved infection ratios of barley plants compared with *dsgfp*-injected SBPHs (Fig. 5i). Immunoblotting analyses showed that protein levels of BYSMV N and RFP in the barley plants fed by *dscsn5*-injected SBPHs increased to more than two-fold in comparison with those of *dsgfp*-injected SBPHs (Fig. 5j). Collectively, these results indicate that knockdown of *LsCSN5* inhibits TIM oscillation and improves insect locomotor activity, phloem feeding duration, and virus transmission capability.

## Heterologous expression of BYSMV P6 in the nervous system of *Drosophila melanogaster* disturbs circadian rhythm and induces high locomotor activity

We next examined whether BYSMV P6 alone could induce abnormal circadian rhythms in insects. Since LsCSN5 and the homolog of *Drosophila* (DmCSN5) share 75.9% identity in amino acid sequence (Supplementary Fig. 6c), we next wonder whether the P6–CSN5–CUL1–TIM module affected the circadian rhythm in *Drosophila*. To test the P6–DmCSN5 interaction, we first cloned the *DmCSN5* gene, purified the DmCSN5-His protein that was then incubated with GST, GST-P6, or GST-P6^II6A. GST pull-down assays showed that DmCSN5-His directly interacted with GST-P6, rather than GST or GST-P6^II6A (Fig. 6a). To achieve heterologous expression of P6 and P6^II6A in *Drosophila*, we first generated transgenic *Drosophila* lines of UAS-P6-GFP and UAS-P6^II6A-GFP, respectively. Then, the transgenic lines were crossed with Elav-Gal4 *Drosophila* line to express P6-GFP or P6^II6A-GFP exclusively in the nervous system (Fig. 6b). Immunoblotting analyses confirmed the expression of P6-GFP and P6^II6A-GFP in the crossed *Drosophila* lines (Supplementary Fig. 8a).

We further used the DAM2 system to monitor the locomotor activity of transgenic *Drosophila* lines under the LD cycle. Inactivated UAS-P6-GFP or UAS-P6^II6A-GFP transgenic flies, as well as expression lines of UAS-P6^II6A-GFP (UAS-P6^II6A-GFP/Elav-Gal4) flies, display strong morning/evening peaks of locomotor activity and normal rhythm in LD cycles (Fig. 6c, d). However, the flies expressing P6-GFP (UAS-P6-GFP/Elav-Gal4) almost lost locomotor rhythm, and their locomotor activity exhibited relatively higher levels than other transgenic flies in all the

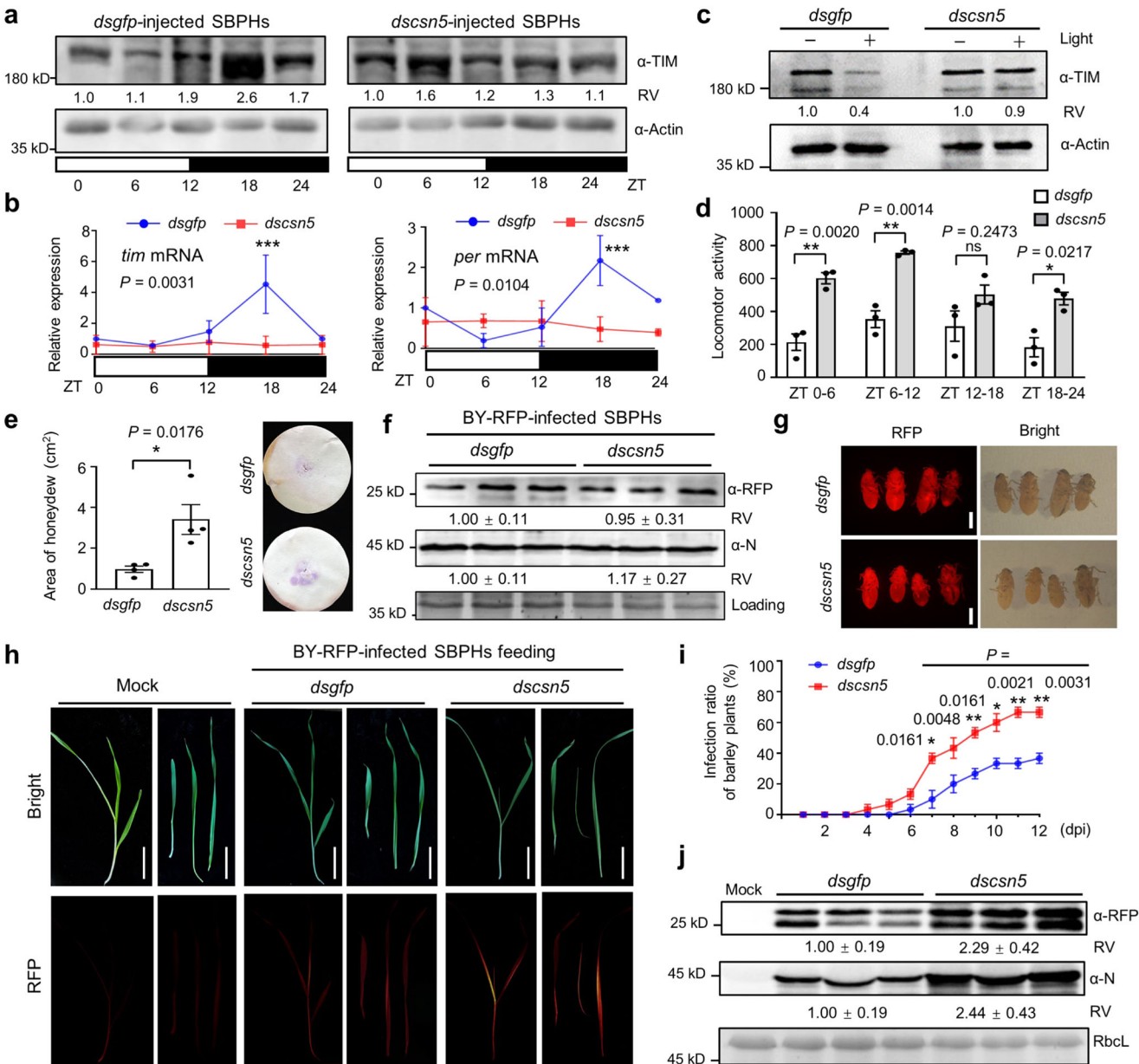

**Fig. 5 | Knockdown of *LsCSN5* inhibits TIM oscillation and improves virus transmission capacity of SBPHs. a** Immunoblotting analyzing accumulation of TIM in SBPHs under LD cycles. Actin served as loading control. Relative values (RV) were calculated from band densities. ZT0 values acted as 1. **b** RT-qPCR analyzing relative levels of *tim* and *per* transcripts in **a**. Accumulation at ZT0 of *dsgfp*-treated SBPHs served as 1. *Actin1* served as endogenous control. Error bars indicates SEM ($n = 3$ independent experiments, 20 SBPHs per repeat). Two-way ANOVA followed by Tukey's test was used to investigate main effects of *dscsn5* on *tim* and *per*. Asterisks indicate significant differences between *dsgfp* and dscsn5-treatment at same points. ***$P < 0.001$. **c** Immunoblotting analyzing accumulation of TIM in SBPHs treated with 0- or 10-min light. Actin served as loading control. **d** Locomotor activity indicated by total activity counts (*y* axis) of 32 insects in 6 h. Error bars indicate SEM ($n = 3$ independent experiments, 32 SBPHs per repeat). Two-way

ANOVA was performed to investigate effects of *dscsn5* on insect activity ($P = 2.7 \times 10^{-7}$). Asterisks indicate significant differences between *dsgfp*- and *dscsn5*-treatment at same points. *$P < 0.05$ and **$P < 0.01$. **e** Area analyses of honeydew excretion by SBPHs ($n = 4$ independent experiments, 10 SBPHs per repeat). **f** Immunoblotting analyzing RFP and N accumulation in SBPHs. SBPHs were infected by BY-RFP for 12 d and injected with *dsgfp* or *dscsn5* for 6 d. **g** RFP fluorescence of SBPHs in **f**. Scale bars, 1 mm. **h** Representative images of barley plants fed by SBPHs in **g**. Scale bars, 5 cm. **i** Infection rates of barley plants in **h** after inoculation ($n = 3$ independent experiments, 10 plants per repeat). **j** Immunoblotting analyses detecting accumulation of N and RFP from barley plants of **h**. RbcL served as loading control. RV represents means ± SEM. **e**, **i** error bars indicate SEM. *$P < 0.05$ and **$P < 0.01$ (two-sided *t* test). **f**–**g** were repeated three times independently with similar results. Source data are provided in the Source Data file.

periods (Fig. 6c, d). Moreover, heterologous expression of P6 in neurons caused 73.5% of the transgenic flies to become arrhythmic, which was significantly higher than the arrhythmic ratio (17.5%) of P6$^{I6A}$, as well as inactivated transgenic lines (arrhythmic ratios, 18.2%, and 14.0%) (Supplementary Fig. 8b). When these flies were incubated in the DD cycle, the rhythmic locomotor pattern was disrupted in the flies expressing P6-GFP (UAS-P6-GFP/Elav-Gal4), but maintained at other transgenic flies, indicating that BYSMV P6 affected endogenous

rhythms (Supplementary Fig. 9). Constant light (LL) treatment abolished the locomotor rhythmicity in these control transgenic flies due to light-triggered TIM degradation (Supplementary Fig. 10). Interestingly, the flies expressing P6-GFP (UAS-P6-GFP/Elav-Gal4) displayed a rhythmic locomotor pattern in LL probably due to the P6-mediated compromised degradation of TIM (Supplementary Fig. 10).

We further determined whether P6-GFP or P6$^{I6A}$-GFP modulates oscillation of *per* and *tim* mRNAs at five time points (ZT0, 6, 12, 18,

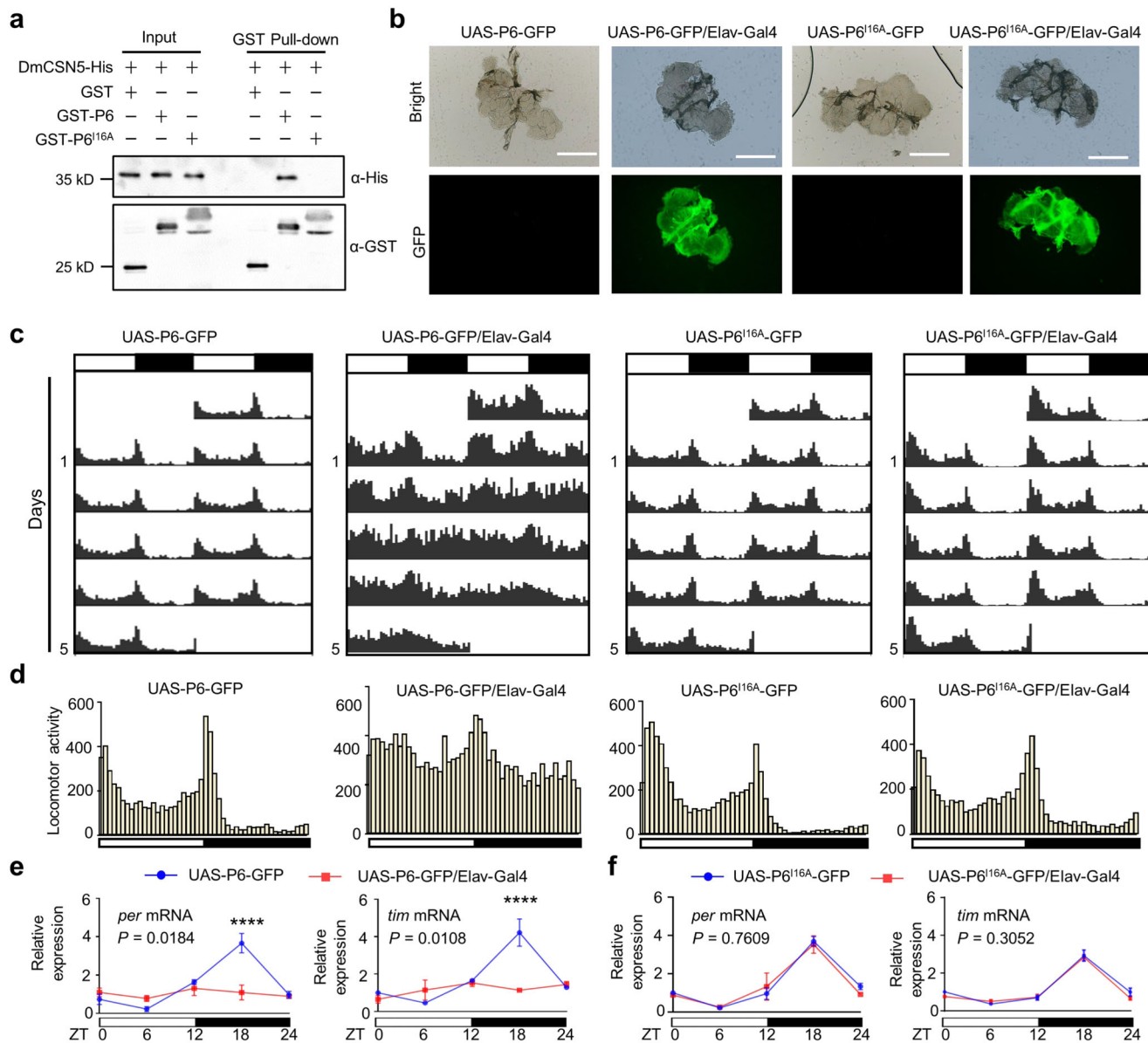

**Fig. 6 | Heterologous expression of BYSMV P6 in neurons of *Drosophila melanogaster* disturbs circadian rhythm and enhances the locomotor activity.**
**a** GST pull-down assays showing DmCSN5–P6 interactions in vitro. DmCSN5-His was incubated with GST-P6, GST-P6^II6A, or GST, and then immunoprecipitated with Glutathione-Sepharose beads for pull-down assays. The experiments were repeated three times independently with similar results. **b** Confocal microscopy images of brain tissues isolated from transgenic fly lines. The GFP fluorescence was from P6-GFP or P6^IIA-GFP. Scale bars, 200 μm. The experiments were repeated three times independently with similar results. **c** Locomotor activity of transgenic flies in LD cycles. The fly genotypes were indicated on top of the panels. The column numbers represent sums of 16 transgenic male flies. White and black bars indicate light and dark phases, respectively. **d** Comparison of the circadian locomotor activity of the indicated transgenic flies. Histograms represent the distribution of activity counts (*y* axis) every 30 min of transgenic lines (16 male flies) through 24 h, averaged over 5 LD days. **e**, **f** RT-qPCR analyzing relative levels of *tim* and *per* transcripts of dissected heads from transgenic fly lines under LD cycles. Accumulation at ZT0 of non-activated P6/P6^IIA (blue) was set as one unit. The *Drosophila tubulin* gene served as an endogenous control. The *Drosophila tubulin* gene served as an endogenous control. Error bars indicate SEM (*n* = 3 biologically independent experiments, 20 insect heads per experiment). Two-way ANOVA followed by Tukey's test was performed to investigate the main effects of P6 *and* P6^IIA expression on mRNA levels of *tim* and *per*. Differences were considered significant at *P* < 0.05. Asterisks indicate significant differences between non-activated P6/P6^IIA (blue) and activated P6/P6^IIA (red) at the same indicated time points. ****P < 0.0001.

and 24). As expected, the amounts of *per* and *tim* mRNAs exhibited a daily oscillation in inactivated transgene lines (UAS-P6-GFP, blue), while oscillation of *per* and *tim* mRNAs were attenuated in heterologous expression flies of P6-GFP (UAS-P6-GFP/Elav-Gal4, red) (Fig. 6e). By contrast, heterologous expression of P6^IIA-GFP had no obvious effect on oscillation of the *per* and *tim* mRNAs (Fig. 6f). Consistently, oscillations of the *clk* and *cyc* mRNAs were also inhibited by heterologous expression of P6-GFP, rather than P6^IIA-GFP (Supplementary Fig. 11).

Collectively, the results demonstrate that BYSMV P6 interacts with DmCSN5, and heterologous expression of BYSMV P6 in neurons disturbs the circadian rhythm and prolongs the high activity of morning and evening peaks.

## Discussion

Many economically important plant viruses are transmitted by insect vectors. The feeding behavior of insects is a key determinant of plant virus transmission. It has long been known but is still at the beginning

of our understanding how viruses induce hyperactive behavior in their insect hosts. Baculoviruses induce hyperactive trait in their caterpillar hosts (order *Lepidoptera*), which is induced by the conserved protein tyrosine phosphatase (*ptp*) gene in a subset of baculoviruses[59]. Virus infection in brain of aggressive bees indicates that virus infection was associated with aggressive trait[13,60]. Since most insect-borne plant viruses do not cause obvious cytopathology, it remains technically challenging to select alive viruliferous insects from healthy controls for comparison of their behavior. To circumvent this problem, we deployed a recombinant BYSMV-RFP (BY-RFP) expressing an *RFP* reporter gene to easily differentiate viruliferous SBPHs from healthy SBPHs, which is crucial for monitoring virus-modulated feeding behavior accurately. As shown in Fig. 1 and Supplementary Fig. 1, we clearly observed fluorescence of BYSMV-expressed RFP in whole bodies of SBPHs, as well as some RFP-labeled CNS of SBPHs (Supplementary Fig. 2, 3), indicating the neurotropic trait of BYSMV. Using this trackable method and selected SBPHs, we obtained unbiased behavioral data and found that BYSMV infection induced SBPH hyperactivity and increased feeding duration to increase virus transmission capacity of SBPHs.

We further explored the molecular mechanisms underlying BYSMV-altered SBPH feeding behavior. A previous study has showed that the COP9 signalosome of *Neurospora* plays important roles in the circadian clock by controlling the stability of the SCF complexes[61]. The null mutations of *Drosophila CSN4* and *CSN5* inhibit light-dependent TIM degradation and clock resetting[27]. Here, the BYSMV accessory protein P6 interacts with LsCSN5 and inhibits its de-neddylation activity on CUL1, which subsequently interferes with TIM degradation and prevents clock resetting in SBPHs (Figs. 3–5). Most insects exhibit locomotor peaks at morning and dusk before sunrise and sunset (morning and evening peaks), which is regulated by endogenous circadian clocks[62]. Thus, we conclude that BYSMV P6-mediated inhibition of TIM degradation prolongs the morning and evening locomotor peaks of SBPHs. These results provide a molecular mechanism to improve our understanding virus-induced feeding behavioral changes in insect vectors.

Many of insect-borne plant viruses can infect both host plants and insect vectors. The COP9 signalosomes were conserved protein complexes in eukaryotic organisms[56]. Indeed, our previous studies have shown that BYSMV P6 can interact with the barley CSN5 and impair the CSN function on JA signaling pathway[49]. Consequently, the JA signaling of infected host plants is dampened, which will thus favor the insect selection and settlement and then indirectly modulate the feeding behavior of insect vectors[49]. Here, we showed that the BYSMV P6 directly alters locomotor rhythm and feeding behavior of insect vectors after virus infection. Therefore, these results demonstrate that BYSMV P6 manipulate insect behavior in both direct and indirect manners for improving virus transmission. However, we currently failed to rescue BYSMV-ΔP6 in its insect vectors to investigate the P6 function in insect behavior. Thus, we assume that P6–LsCSN5 interaction inhibits the CUL1-based E3 ligase-mediated degradation of other targets than the TIM protein, which would contribute to other important functions of BYSMV P6 in virus infections. Nonetheless, we found that transgenic *Drosophila* lines of BYSMV P6 were impaired in the circadian rhythm and became more active than non-activated line or P6[I16A] transgenic lines (Fig. 6). These results imply that BYSMV P6 targets different CSN5 orthologues of insects and alters their circadian rhythm.

In summary, we used multiple approaches to explore how an insect-borne rhabdovirus modifies the feeding behavior of its insect vectors. The *per/tim* transcriptional feedback loop of insect circadian clock, involving PER/TIM and CLK/CYC, controls rhythmic behavior of insects. The CSN5 protein is a regulator in mediating ubiquitination and proteasomal degradation of TIM to reset the clock. BYSMV, an important insect-borne cytorhabdovirus, produces an accessory P6

protein interacting with CSN5 and inhibiting its function on the degradation of TIM. Consequently, BYSMV infection inhibits the transcriptional feedback loop of insect circadian clock, leading to hyperactivity for enhancing virus transmission (Fig. 7).

Considering that animals infected by RABV exhibit rabies signs of abnormally aggressive behavior[30–33], it is not surprising that plant rhabdoviruses can drive the insect vectors into hyperactivity for virus spread. Many rhabdoviruses can manipulate host biting/feeding behavior in human, animal, and plant hosts. However, since other rhabdoviruses do not encode homologs of BYSMV P6, it remains to be determined which components of these rhabdoviruses are responsible for modulating host behavior. Nonetheless, our results together with our previous studies[49] demonstrate that BYSMV P6-mediated modification of host plants and insect vector represents a new layer of co-evolutionary arms race between rhabdoviruses and their hosts.

## Methods

### Virus inoculation and insect rearing

The recombinant BYSMV-RFP (BY-RFP) virus was maintained in barley plants (*Hordeum vulgare* cv Golden promise) through transmission by the small brown planthoppers (SBPHs, *Laodelphax striatellus*) as described previously[50]. Healthy and viruliferous SBPHs were reared on rice seedlings in chambers at 25°C with 12 h: 12 h (light: dark) photoperiod. Flies were raised and crossed at 25°C on standard cornmeal agar media. Transgenic flies were generated in the background of *Drosophila* strain $w^{1118}$ using the GAL4/UAS system. We cloned the ORFs of P6 or P6[I16A] into the pUAS vector, in which the ORFs were fused between upstream activating sequence (UAS) of yeast transcription factor GAL4 and GFP. The UAS-P6-GFP and UAS-P6[I16A]-GFP transgenic flies were generated by the Tsinghua Fly Center (Beijing, China). Then the transgenic male flies were crossed with the virgin flies with Elav-Gal4 driver to generate heterologous expression flies of P6-GFP or P6[I16A]-GFP in the nervous systems.

### Histological section staining and image acquisition

Viruliferous and healthy SBPHs were fixed in 4% paraformaldehyde overnight at 4°C followed by removal of all the legs. Samples were dehydrated in a series of ethanol, cleared in xylene, embedded in paraffin. The samples were sectioned serially at 8 μm in a microtome and collected on adhesion microscope slides as described previously[63]. After deparaffinization and re-hydration, samples were stained with DAPI and observed by an Andor Dragonfly spinning-disc confocal microscope. For high-resolution detections in longitudinal sections of whole bodies, localization of RFP and DAPI-stained nuclei was monitored in a z-stack model with an optical slice thickness of 3 μm, and confocal sections covered ~8 μm sample thickness using a ×10 0.32 N.A. objective (Leica HC PL FLUOTAR). The detailed localization in brains was acquired with an optical slice thickness of 0.3 μm using a ×100, 1.44 N.A. oil objective (Leica HC PL APO). The signals were recorded by a camera (Andor Zyla 4.2) of scientific complementary metal-oxide semiconductor (sCMOS). DAPI and RFP were visualized at excitation of 405- and 568- nm, respectively. In detail, images were acquired with 405-nm (30–35%), exposure time 300 milliseconds (ms) and 568 nm (20–25%), exposure time 100–200 ms. Images were acquired by the Fusion 2.1 software (https://andor.oxinst.com/products/dragonfly#fusion), and processed by the Imaris software (Bitplane AG, Zurich, Switzerland). The positions of SBPH neuropil were according to the three-dimensional reconstruction of planthoppers as described previously[64].

### Fluorescence image acquisition

Whole bodies of Mock or BY-RFP-infected SBPHs were photographed with an Olympus stereomicroscope SEX16 (Olympus, Tokyo, Japan). Dissected brains of SBPHs and transgenic flies were photographed with the BX53 microscope using the DP72 digital

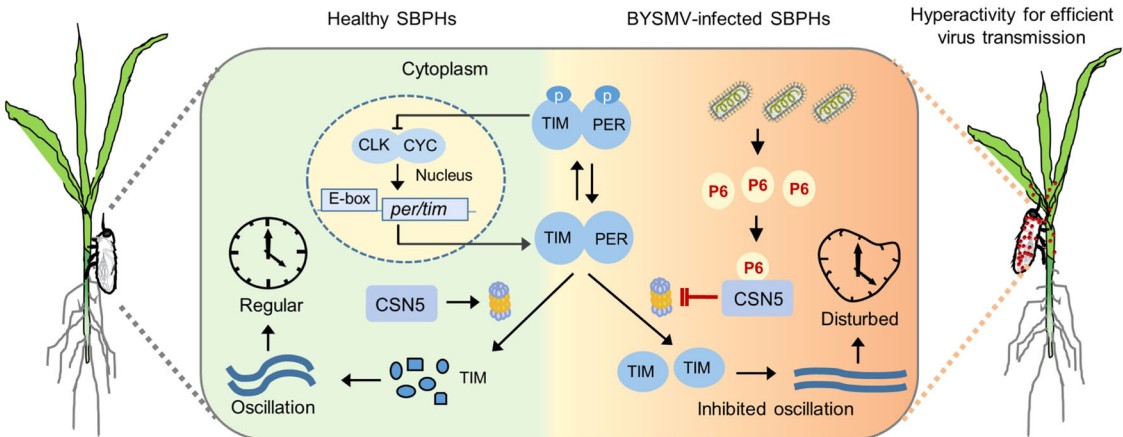

**Fig. 7 | Proposed model of BYSMV-modulated locomotor activity of insect vectors to enhance viral transmission.** In healthy insects, two transcription factors, CLOCK (CLK) and CYCLE (CYC), form a heterodimer and promote transcription of *period* (*per*) and *timeless* (*tim*) genes. Subsequently, the PER and TIM proteins gradually accumulate and are phosphorylated. Then, PER and TIM form a heterodimer and move into the nucleus to repress the transcription activity of CLK/ CYC. Subsequently, TIM is subjected to ubiquitination and proteasomal degradation through the CSN5-regulated CUL1-based E3 ligases, leading to clock resetting. These processes form a *PER/TIM* oscillatory loop and regular locomotor rhythms. By contrast, BYSMV infection produces the accessory protein P6 that directly targets to CSN5 and inhibits its functions in TIM degradation. Finally, the oscillatory of TIM is inhibited and the locomotor rhythms are disturbed, which results in hyperactivity of insect vectors for virus transmission.

camera (Olympus, Tokyo, Japan). Some dissected brains were observed with a Leica TCS-SP8 laser scanning confocal microscope. For Immunological detection of N protein in brain tissues of SBPHs, Heads of non-inoculated and BY-RFP-infected SBPHs at 12 dpi were dissected and placed in 4% paraformaldehyde for fixation, followed by the added of rabbit antibodies to the N protein, and incubation with alex-488-coupled rabbit secondary antibody (GFP), observed under the BX53 microscope. GFP and RFP were visualized, respectively, at excitation of 488 and 543 nm and detected at 500–540, 585–625 nm, respectively.

### Locomotor activity analysis

Locomotor activity of planthoppers or flies was detected using the Drosophila Activity Monitor 2 (DAM2) system (TriKinetics, Waltham, MA) as described previously[27,65]. Briefly, the assays were performed in a chamber with a condition at $25 \pm 2\,°C$ and 50% humidity. Insects were placed into glass tubes individually and exposed to a 12 h: 12 h (Light: Dark) LD cycles for 5 d. The light onset was taken as Zeitgeber time 0 h (ZT0). The infrared beam passing through the middle of tubes was interrupted by insect movement and was automatically recorded every 30-min. The results of locomotor activity were analyzed using the ActogramJ software in actogram format[66]. The FaasX software was used for analysis of fly rhythm[67]. Rhythmic flies were defined with the following criteria of $\chi^2$ periodogram analysis: power ≥20 and width ≥1.5. The constant darkness (DD) or light (LL) was settled for more than 5 d following 2-d LD cycles.

### RT-qPCR

RT-qPCR assays were performed as described previously[68]. Briefly, total RNA was extracted using TRIzol reagent (Sigma-Aldrich, USA) and treated with *DNase* I (TaKaRa, Beijing, China) to remove contaminated genome DNA. Next, 1–2 µg total RNA was used for reverse transcription with oligo (dT) using HiScript® II Reverse Transcriptase (Vazyme, Nanjing, China). Quantitative PCR was carried out on the Bio-Rad CFX96 Real-Time PCR System using 2× SsoFast EvaGreen Supermix (Bio-Rad, Beijing, China). SBPH *β-actin1* and *Drosophila tubulin* served as endogenous controls. Three independent experiments were carried out for biological statistics. All primers were listed in Supplementary Table 1.

### Immunoblotting analysis

To generate anti-LsCUL1 and anti-LsTIM antibodies, the cDNA fragments corresponding to the *CUL1* fragment (aa 354-813) and *TIM* fragment (aa 1-338) of SBPHs were introduced into the pET-30a (+) vector, respectively. The resultant plasmids were transformed into the *E. coli* strain BL21 for expression of LsCUL1-6×His and LsTIM-6×His protein. The 6×His-tagged proteins were purified using the Ni-NTA agarose (Bio-Rad, USA, 780-0801) and used to immunize rabbits for anti-LsCUL1 and anti-LsTIM antibodies (Beijing Genomics institution, Beijing, China). Immunoblotting analysis was carried out as previously[69]. Briefly, total proteins were extracted from SBPHs in RIPA Lysis Buffer with phenylmethylsulfonyl fluoride (PMSF) using a TGrinder high-speed tissue grinder (Tiangen Biotech, Beijing, China). After incubating on ice for 30 min, the homogenate was centrifuged at $13,000 \times g$ for 5 min at 4°C. The supernatant was separated in SDS-PAGE gels and transferred to nitrocellulose membranes. Membranes were blocked with 5% skim milk and incubated with the polyclonal antiserum of the BYSMV N (1:3000), P6 (1:3000), RFP (1:2000), GFP (1:3000), GST (1:5000), His (1:5000), LsCUL1 (1:1000), Actin (1:3000), or LsTIM (1:1000) antibodies at 37°C for 1 h. Goat anti-rabbit or anti-mouse IgG horseradish peroxidase conjugate (1:30,000) were used as secondary antibodies. Finally, membranes were incubated with NcmECL Ultra chemiluminescent substrate (Cat. No. P10300, New Cell & Molecular Biotech Co., Ltd, China) and analyzed using Azure Sapphire RGBNIR (Azure Biosystems, USA).

### EPG assays

EPG assays were performed as described previously[53]. Feeding behaviors of SPBHs on barley were monitored continuously for 8 h using an 8-channel DC-EPG device (EPG systems, Wageningen University, The Netherlands) in an electrically grounded Faraday cage. After a 4 h pre-starvation period, dorsum of healthy and infected planthoppers was connected to a gold wire (2-cm long and 12.5-µm in diameter) through a water-soluble silver conductive paint (Colloidal Silver; Ted Pella, Inc.). The other end of the wire was connected to the Giga-8 DC-EPG amplifier through the EPG probe. Then the planthoppers were carefully moved to barley plants. The hard copper wire of plant electrode was inserted into soil. The data sets were recorded and analyzed by Stylet+ software (Wageningen University, The Netherlands).

## Yeast two-hybrid assays

To verify the P6–LsCSN5 interaction, the ORFs of P6 and P6$^{II6A}$ were individually cloned into the pGADT7 vector as described previously[49], and the LsCSN5 ORF was introduced into the pGBKT7 vector. Plasmids expressing AD-P6/BK-LsCSN5, AD-P6$^{II6A}$/BK-LsCSN5, and negative controls were transformed into AH109 or Y187 yeast cells and grown in Trp/Leu double-deficiency yeast plates. The co-transformed cells were diluted and dropped on -Trp/-Leu double-deficiency and -Trp/-Leu/-His/-Ade/3-AT quadruple-deficiency yeast plates with 5 mM 3-amino-1,2,4-triazole (3-AT) and incubated at 30°C for 3–5 d.

## GST pull-down assay

The ORFs of LsCSN5 and DmCSN5 were cloned into the pET-30a (+) vector for expressing LsCSN5-6×His and DmCSN5-6×His proteins, respectively. The ORFs of P6 and P6$^{II6A}$ were engineered into the pGEX-KG vector for expressing GST-P6 and GST-P6$^{II6A}$, respectively. Fusion proteins were expressed in the *E. coli* strain BL21 under induction of 0.1 mM Isopropyl-D1-Thiogalactopyranoside at 18°C for 16 h. In GST pull-down assays, GST, GST-P6, or GST-P6$^{II6A}$ proteins were incubated with LsCSN5-6×His or DmCSN5-6×His proteins in 500 μL binding buffer (50 mM Tris-HCl, pH7.5, 250 mM NaCl, 0.6% TritonX-100, 0.1% glycerol, 1× cocktail, 5 mM DTT) with 30 μL glutathione Sepharose4B beads (GEHealthcare, USA) at 4°C for 2 h. After centrifugation at $800 \times g$ for 1 min, beads were washed five times with series concentrations of washing buffer (50 mM Tris-HCl pH7.5, 150–250 mM NaCl, 0.6% TritonX-100, 1× cocktail), boiled in SDS buffer for immunoblotting analyses with anti-GST (1:5000) and anti-His (1:5000) antibodies.

## Co-immunoprecipitation (Co-IP) assay

Co-IP analysis was performed as described recently[69]. To pull down the P6 protein from viruliferous SBPHs, total proteins were extracted from 4th-instar viruliferous planthoppers using RIPA Lysis Buffer (Strong) supplemented with PMSF. Approximately 10% of the total protein fraction served as input samples. Then, the extracted total protein samples were incubated with LsCSN5-6×His was at 4°C for 3 h, and then added Ni-NTA agarose for 2 h. The GFP-6×His protein was used as a negative control. Finally, the beads were washed and boiled for immunoblotting analyses.

## Knockdown gene by dsRNA injected

The specific fragments of *LsCSN5* (nt 175-588, ORF) were amplified using primers containing T7 RNA polymerase promoter at the two termini as shown in Supplementary Table 1. The purified PCR products were used as templates for dsRNA synthesis using T7 RiboMAX Express RNAi System kit (Promega). Transcripts of *dsGFP* served as a negative control. Approximately 40 ng dsRNA (2.0 μg/μL) was injected into the thorax of SBPHs.

## Honeydew measure

To detect SBPH honeydew secretion, barley plants (one plant per cup) were covered with a transparent plastic cup and placed over a filter paper. Ten SBPHs were placed into the cups for feeding. After 2 d, the filter papers were dried in an oven for 30 min, and then soaked with 0.1% (w/v) ninhydrin in acetone solution. After 30 min at 65°C, the amino acid contents of honeydew were stained until violet or purple spots appeared. The areas of stained honeydew were analyzed by Image J software.

## Data analysis

Statistical analyses were performed using Graphpad Prism 8. Two-tailed Student's *t* test was used to compare the significance of only two groups. Statistical analyses of RT-qPCR of clock-related gene mRNAs in different SBPHs among Zeitgeber times were performed using two-way ANOVA followed by Tukey's post hoc test, $P < 0.05$. Asterisks indicate statistical significance (*$P < 0.05$, **$P < 0.01$, ***$P < 0.001$, ****$P < 0.0001$).

## Accession numbers

The accession numbers of described genes in this study can be found in GenBank database (https://www.ncbi.nlm.nih.gov/) with the following accession numbers: *Lstimeless* (MG356485.1); *Lsperiod* (MG356486.1); *DmCSN5* (NM_058094.5); *LsActin* (AY192151); *Dmtimeless* (AB059649.1); *Dmperiod* (NM_080317.2); *Dmclk* (NM_079240.3); *Dmcyc* (NM_079444.3); *Dmtubulin* (M14643.1); *Lsclk* (OR514639); *Lscyc* (OR514641); *Lscsn5* (OR514640); *LsCUL1* (OR514638).

## Reporting summary

Further information on research design is available in the Nature Portfolio Reporting Summary linked to this article.

## Data availability

The data supporting the findings of this study are available within the Figures and Supplementary information. The source data for Fig. 1–6, and Supplementary Figs. 1, 4, 7–11 are provided as a Source Data file. Source data are provided with this paper.

## Code availability

The stylet+ software for EPG experiment is freely available (www.epgsystems.eu). The locomotor data were analyzed by FaasX[67] (Fly Activity Analysis Suite) on a Macintosh computer (https://neuropsi.cnrs.fr/en/departments/cnn/group-leader-francois-rouyer/) and the ActogramJ software[66] (https://bene51.github.io/ActogramJ/) can be downloaded freely.

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

## Acknowledgements

We thank our colleagues Jialin Yu, Dawei Li, and Yongliang Zhang for their helpful discussion. We thank Professor Juan Du (China Agricultural University) for constructive suggestions and technical assistance in DAM2 analysis. We thank Professor Lili Zhang and Dr. Yan Huo (Institute of Microbiology, Chinese Academy of Sciences), as well as Professor Xun Zhu (Institute of Plant Protection, Chinese Academy of Agricultural Sciences) for their technical assistance in EPG assays. This work was funded by the National Natural Science Foundation of China (32370154 to X.B.W., 32102150 to Q.G.) and the Chinese Universities Scientific Fund (2023TC154 to X.B.W.).

## Author contributions

D.M.G. performed the majority of the experiments, assisted by J.H.Q., J.W.Z., Y.Z., Q.G., L.X., and Y.Z. D.M.G., X.B.W., Y.W., J.F., H.Z., and C.G.H. analyzed the data. X.B.W. and D.M.G. wrote the manuscript. All authors discussed the results and commented on the manuscript.

## Competing interests

All authors declare no competing interests.
