## [Peer Review File · Nature Communications]

Reviewers' Comments:

Reviewer #1:

Remarks to the Author:

This is a very interesting ms that suggests that the virus from plants mediates hyperactivity and increased feeding in planthoppers which in turn further propagates the virus. The authors suggest that the circadian clock is disrupted although to this reader at least, they do not convincingly demonstrate this. They may be right but the ms needs to be tightened up considerably with some additional experiments.

L52-56....aggressive bees, food intake? Please suggest ways in which these phenotypes enhance virus transmission.

L70 refs 25 and 26 refer to *Drosophila* and *Neurospora* respectively. The way it is reported it suggests to the reader that these cullin interactions are found in flies.

L70-72 Ref 27 has nothing to do with the previous sentence, other than being a very general statement. It does not naturally follow

L108 replace 'healthy' with 'non-inoculated control'

L110 replace 'neuron' with 'nervous'

L119-129 – sleep analysis might be useful here but data collected in 30 min bins – shame

Fig 1e please put in Supps a typical activity trace of one animal from infected and not infected animals.

L150 ninhydrin – why ? used for fingerprints and ammonia detection usually?

L163 the *per* and *tim* transcripts should be annotated in lower case italics in text and figures

Fig 3 & L167 'consistently' implies replicate gels. I think only one western was performed for each experiment

Fig 3a 3b. because we assume a negative feedback model, lower *per* means higher *tim* and vice-versa. As both *per* and *tim* transcripts are affected ~ equally and reduced, does this not suggest that the primary effect could be on the Clock/Cyc TFs? Also ANOVA followed by posthoc testing is better than multiple t-tests for all qPCR results.

Fig 3c. poor westerns, overexposed. Where are the MW markers for actin and TIM? I assume this is the whole animal, and only one blot, no replicates? Need at least 3 replicates as with the mRNA plots. Also I think the days of simply showing the cutaway bands in westerns are over, so show the whole blot in a supplementary. Was the membrane incubated with both actin and TIM Abs, or stripped then incubated with actin Ab or were the samples run 2x and incubated with each Ab- Methods does not say. Also because of the overexposure, maybe each lane of TIM should be normalised against total protein in each lane. Given there is only one blot, perhaps the RV numbers should simply reflect the ratio of TIM/actin. In that way the reader can see what the differences are between healthy and inoculated hoppers – we do not need a further transformation to ZT24 value. That would be necessary only if some stats were to be done using biological replicates.

Fig 3d was light treatment immediately after ZT24? What does 10, 30 min treatment mean? Is this a 10 min exposure after ZT24 and a 30 min exposure? – again no Methods statement.

L177-8 As there is no constant darkness experiment the authors do not know whether the circadian clock has been compromised. The authors need to perform a DD and an LL experiment. The ms suggests that only the light phase of TIM degradation is affected, so DD rhythms may not be affected, which would confirm that the effects are light specific. But we might also expect a circadian rhythm to continue running in constant light, LL if TIM degradation is compromised, perhaps with a longer period as *tim* RNA is still cycling in inoculated hoppers? In any case both experiments would be informative and should be done.

L213 *dscsn5* comes out of nowhere. Please state that this is a ds RNAi interference. There are a lot of acronyms in the ms and this caused some confusion to this reader.

Fig 5A, overexposed westerns –again replicates are required. The authors need to show that the low levels of *tim* mRNA at night in inoculated animals are due to higher levels of TIM at night (more RNA repression, negative feedback). Again the *per* results are not consistent with a negative feedback model.

Fig 5b – 2 way ANOVA not multiple t-testing

Fig 5c – lack of replicates. Absolute minimum of two required, three to generate an sem.

Fig 5f, 5g – quantification?

L248 dpi? days post infection?

Fig 6 title 'enhances' rather than 'improves'

Fig 6e ANOVA

The locomotor data in Fig 6 also need to be collected in DD and LL to examine any effects on clock function. Simply observing elevated locomotor levels may not be anything to do with the clock itself – maybe just the locomotor output is affected

Supp Fig 6a not necessary

L280 The authors claim high levels of arrhythmicity in the overexpressed P6 flies but do not give their criteria for arrhythmicity. Was this simply by eyeballing the LD patterns? A spectral analysis of the data is required.

In conclusion, while the results are interesting, I do not see the connection between hyperactivity, increased feeding and clock dysfunction as proven. Perhaps the products of locomotor activity output genes are targeted directly by COP9 mediated degradation? Furthermore, many of the results need replication of westerns particularly with associated statistical analyses and locomotor experiments to be performed in constant conditions, DD and LL. The results also do not conform to predictions based on the negative feedback model of insect clocks. There is work by Sehgal's group showing that Clock mutants also show hyperactivity. If CLOCK is targeted by the virus, this could explain why both *per* and *tim* transcripts are affected equally in inoculations.

Reviewer #2:

Remarks to the Author:

Here the authors use a RFP-tagged BYSMV and localize it in intact insects (Fig. S1) and in tissue sections (Fig. 1b). From Fig. S1a it seems that the RFP label is homogeneous all over the insect, whereas in Fig. 1b it is rather punctuate and there is a big RFP-labeled aggregate in the lower left corner, whereas the head is on the right side and contains less label. Fig 1c shows RFP label in the brain of dissected heads, but not in salivary glands. Why is this so? Has the virus not yet invaded the salivary glands at 12 days? On the other hand, there is a strong label, apparently stronger than in the brain, in a structure next to the gland. Please discuss this. Then locomotor activity is assessed by DAM2 and shows clearly an increase in activity of infected insects. I do not understand though, what the individual columns along the timeline (1 to 5 is for 1 to 5 days, I assume) in Fig. 1e present exactly (I assume it is some kind of movement activity, but I did not find the information neither in materials and methods, nor in the figure because there is no labeling (neither unit nor parameter) of the y-axis. Then the feeding behavior is measured by EPG and honeydew assay and shows increased phloem feeding of infected insects, compatible with increased virus inoculation. However, it is unclear whether the planthoppers were placed on healthy or infected plants for EPG and honeydew analysis. Please add this relevant information. RT-qPCR and western blotting analysis indicates that TIM and PER oscillation is impaired in infected insects. This could be due to interaction of BYSMV protein P6 with planthopper protein CSN5, which is then tested and verified by several interaction assays. Evidence for a functional role of this interaction is shown by the fact that P6 inhibits CSN5-dependent deneddylation of CUL1, a component of the proteasome degradation machinery. Deneddylation is required for degradation activity, therefore P6 could inhibit CUL1-dependent degradation-mediated oscillation of TIM and PER levels by interacting with CSN5. The authors present evidence for this by silencing CSN5 and measuring TIM and PER levels. Further they show that this correlates with increased locomotion and transmission. Finally, P6 expression in *Drosophila* has similar effects on locomotion activity, TIM and PER accumulation as in planthoppers, further supporting the authors' hypothesis on P6 action.

Similar results of behavior changes of virus-infected planthoppers and other plant virus vectors have been reported quite often. Relating it with specific changes in the vector has only been reported rarely. This makes this paper really interesting and the authors present convincing evidence that viral P6 interacts with a component of the proteasome degradation pathway and inhibits degradation of circadian oscillation-relevant TIM/PER, thus provoking the behavior changes relevant for more efficient virus transmission.

What is less clear, is the microscopic data. Here the methods are, in my opinion, not sufficiently well described: microscopy settings and acquisition conditions (stacks or optical sections, filter settings etc.) are not mentioned; most acquisitions seem to show images of whole body or dissected organs, but in materials and methods there is also mentioning of tissue sections. I

assume this applies to Fig. 1b, but you should indicate which images show sections. In Fig. 1c and S2, the label is presented in reddish-yellow. Why is this label in yellow, if it is confocal RFP? Are these confocal false color presentations, confocal superpositions of different acquisition channels or are these multicolor epifluorescence acquisitions? Please resolve this issue and describe more precisely the acquisition conditions. Also, the claims of RFP localization seem not always be fully supported by the shown images (see above). The use of free RFP as a marker might result in misleading localization information of the virus, because it can freely diffuse. So while this approach allows clearly to distinguish between infected and healthy planthoppers, precise localization might be difficult to assess. The authors should consider this.

The authors should also describe more in detail the recombinant virus, in particular the RFP insertion; the information in Fig. 1a is not sufficient.

Concerning the discussion, there is some data that other plant viruses modify the nervous system of insect vectors (for example PMID:32729829), and that also other mechanisms might apply to induce transmission-friendly behaviors (for example PMID:35246603). Another point you might want to address: since P6 interacts with CSN5 and indirectly with CUL1, could it also inhibit degradation of other targets than TIM/PER?

Other comments

L 104 Wording a bit strange 'alimentary canals'

L 105 Please indicate how the virus was RFP-tagged. From Fig. 1A it seems that it is free RFP, but please confirm. I did not find information on cloning in material and methods.

L 128 It rather increases than induces activity.

L 276 'Files' should read 'flies'.

Fig. 1b: Bright field images might be useful to illustrate virus localization.

Fig. 6b: Fluorescent imaging of whole flies might be nice to show exclusive brain localization of P6.

Firstly, we would like to thank the Editor and Reviewers for all their efforts in assessing our manuscript and providing helpful and constructive suggestions. We have tried to address these comments through performing additional experiments and modifying statements. The manuscript is indeed much improved. Comments by the reviewers are shown in black, and our point-by-point response is shown in blue.

REVIEWER COMMENTS

Reviewer #1 (Remarks to the Author):

This is a very interesting ms that suggests that the virus from plants mediates hyperactivity and increased feeding in planthoppers which in turn further propagates the virus. The authors suggest that the circadian clock is disrupted although to this reader at least, they do not convincingly demonstrate this. They may be right but the ms needs to be tightened up considerably with some additional experiments.

Response: We appreciate and agree with these comments. As you will see below, we followed your suggestion and performed additional experiments which certainly improves our manuscript.

L52-56.... aggressive bees, food intake? Please suggest ways in which these phenotypes enhance virus transmission.

Response: Many thanks for this comment which allows us to clarify this point. In this paragraph, we cited several previous reports to demonstrate neuroparasitology (Line xx). In these reports, parasite infections can modify behaviors of insect vector for transmission. To clearly explain the phenotypes, we added the comment as following “Most insect-borne plant viruses are transmitted along with salivary secretion during insect feeding on host plants. Therefore, longer feeding time and more food intake can enhance virus transmission probability”. Please see line 56-58.

L70 refs 25 and 26 refer to Drosophila and Neurospora respectively. The way it is reported it suggests to the reader that these cullin interactions are found in flies.

Response: Agree. In rule out this misleading, we removed the Neurospora results of ref 26 to discussion section. **Please see line 79.**

L70-72 Ref 27 has nothing to do with the previous sentence, other than being a very general statement. It does not naturally follow

Response: We agree with the comment and replaced the ref 27 with a new reference (Keegan et al., 2007). Here, we wanted to indicate that endogenous circadian clocks form a transcriptional loop regulating various genes into rhythmic expression. Then, insect behavior and physiological functions are subsequently changed along with circadian clocks. We rewrote the sentence as following: The *PER/TIM* oscillatory loop triggers a rhythmic expression of various clock-related genes for modulation of insect behavior and physiological functions. **Please see line 81.**

References: Keegan KP, et al., Meta-analysis of Drosophila circadian microarray studies identifies a novel set of rhythmically expressed genes. PLoS Comput Biol 3, e208 (2007).

L108 replace ‘healthy’ with ‘non-inoculated control’

Response: Replaced as suggestion. Thanks! **Please see lines 119, 139, 142.**

L110 replace ‘neuron’ with ‘nervous’

Response: Replaced as suggestion. Thanks! **Please see line 122.**

L119-129 – sleep analysis might be useful here but data collected in 30 min bins – shame

Response: We appreciate the valuable suggestion. Usually, DAM2 data was summed into 1 to 5 minute bins when analyzing sleep/rest parameters. However, we wish to point out that our studies mainly focus on the feeding behavior and locomotor activity of different planthoppers. Usually, the DAM2 data was collected in 30 min bins for circadian parameters as previous reports (Jiang et al., 2018; Chiu et al., 2010).

Reference: Jiang Y.D., et al. (2018). Knockdown of timeless disrupts the circadian behavioral rhythms in *Laodelphax striatellus* (Hemiptera: Delphacidae). *Environ Entomol*, 47, 1216–1225.

Chiu, J.C., et al., (2010). Assaying locomotor activity to study circadian rhythms and sleep parameters in *Drosophila*. *Journal of visualized experiments, JoVE* 43, 2157.

Fig 1e please put in Supps a typical activity trace of one animal from infected and not infected animals.

Response: As suggested, we have provided histograms represent the distribution of activity counts (y-axis) every 30 min of indicated SBPHs (n = 32) through 24 h. **Please see the new Fig 1f.**

L150 ninhydrin – why? used for fingerprints and ammonia detection usually?

Response: The honeydew excretion on filter paper can be stained by ninhydrin to indicate the amino acid content, which has been used as an effective and measurable indicator of planthopper feeding amount in previous studies.

References:

Shangguan, X., et al. (2018). A mucin-like protein of planthopper is required for feeding and induces immunity response in plants. *Plant physio.* 176, 552–565.

Du, B., et al. (2009). Identification and characterization of Bph14, a gene conferring resistance to brown planthopper in rice. *PNAS*, 106(52), 22163–22168.

L163 the *per* and *tim* transcripts should be annotated in lower case italics in text and figures

Response: Corrected throughout the manuscript, thanks. **Please see lines 179-185.**

Fig 3 & L167 ‘consistently’ implies replicate gels. I think only one western was performed for each experiment.

Response: We have three independent biological repeats with similar results. We provided these repeats in the source data as requested by the journal. Please see the source data.

Fig 3a 3b. because we assume a negative feedback model, lower per means higher tim and vice-versa. As both per and tim transcripts are affected ~ equally and reduced, does this not suggest that the primary effect could be on the Clock/Cyc TFs? Also ANOVA followed by post hoc testing is better than multiple t-tests for all qPCR results.

Response: We appreciate the helpful suggestion. The *per* and *tim* transcripts usually change simultaneously, which forms a negative feedback with their protein levels and Clock/Cyc TFs. Please see the proposed feedback loop in the following review paper.

Figure 1. The transcriptional feedback loop of the *Drosophila* circadian clock.

Reference: Tataroglu O, Emery P (2014)

Studying circadian rhythms in *Drosophila melanogaster*. *Methods* 68: 140-150.

<https://doi.org/10.1016/j.ymeth.2014.01.001>

In addition, we also tested *clk/cyc* transcripts and both of them lost rhythm changes after virus infection. We analysed all the qPCR results using two-way ANOVA followed by Tukey's test to investigate the main effects of BY-RFP infections on mRNA levels of *tim* and *per*. Differences were considered significant at $P < 0.05$. Asterisks indicate significant differences between mock-treated and BY-RFP-infected SBPHs at the same indicated time points. * $P < 0.05$.

Please see the Page 31, the Fig 3, the figure legends, and the revised methods (Line 544-546).

Fig 3c. poor westerns, overexposed. Where are the MW markers for actin and TIM? I assume this is the whole animal, and only one blot, no replicates? Need at least 3 replicates as with the mRNA plots. Also I think the days of simply showing the cutaway

bands in westerns are over, so show the whole blot in a supplementary. Was the membrane incubated with both actin and TIM Abs, or stripped then incubated with actin Ab or were the samples run 2x and incubated with each Ab– Methods does not say. Also because of the overexposure, maybe each lane of TIM should be normalised against total protein in each lane. Given there is only one blot, perhaps the RV numbers should simply reflect the ratio of TIM/actin. In that way the reader can see what the differences are between healthy and inoculated hoppers – we do not need a further transformation to ZT24 value. That would be necessary only if some stats were to be done using biological replicates.

Response: We appreciate your valuable suggestions.

1: In this revision, we provided the low exposed results.

2: We added the MW markers for actin and TIM in Fig 3c. At the same time, we added MW markers in all other figures.

3: All these western blotting and whole membranes have three biological repeats as shown in the source data as requested by the journal.

4: The protein samples run 2 times with same loading amount and incubated with different antibodies separately.

5: Relative values (RV) of TIM were calculated from band densities and normalize to against Actin accumulation (TIM/Actin). We have updated the figure legends. Please see xx.

6: ZT24 represents the samples collected at 5 min before light. ZT0 represents the samples collected at 5 min after light treatment. Therefore, some difference between ZT0 and ZT24.

Fig 3d was light treatment immediately after ZT24? What does 10, 30 min treatment mean? Is this a 10 min exposure after ZT24 and a 30 min exposure? – again no Methods statement.

Response: Thank you for your questions and we are sorry for the lack of clarity regarding the light treatment. In Fig 3d, the SBPHs at ZT15 (in the middle of Dark) were transferred to light for 0-, 10- or 30- min, and was collected for protein detection. We have added the experiment details in the figure legend. Please see Lines 193 and 816.

L177-8 As there is no constant darkness experiment the authors do not know whether the circadian clock has been compromised. The authors need to perform a DD and an LL experiment. The ms suggests that only the light phase of TIM degradation is affected, so DD rhythms may not be affected, which would confirm that the effects are light specific. But we might also expect a circadian rhythm to continue running in constant light, LL if TIM degradation is compromised, perhaps with a longer period as tim RNA is still cycling in inoculated hoppers? In any case both experiments would be informative and should be done.

Response: We agree with the reviewer and appreciate the valuable suggestion. We performed DD and an LL experiment for SPBHs. Indeed, even in DD cycle, the TIM is still degraded through the CUL1-based E3 ligase. Therefore, the P6-CSN5 interaction affected the locomotor activity of insect in DD cycle. As expected, viruliferous SBPHs still exhibited higher locomotor activity than non-inoculated SBPHs in DD. Whereas, there was no obvious difference between the two kinds of SBPHs in LL. Please see the new Supplementary Fig. S4.

L213 *dscsn5* comes out of nowhere. Please state that this is a ds RNAi interference. There are a lot of acronyms in the ms and this caused some confusion to this reader.

Response: Many thanks for your suggestion. We added the statement as following “To examine whether *LsCSN5* was required for de-neddylation of CUL1, *LsCSN5* dsRNA (*dscsn5*) were synthesized *in vitro* and injected into SBPHs to interfere with endogenous *LsCSN5*, and the *gfp* dsRNA (*dsgfp*) served as a negative control”. Please see Line 233-236.

Fig 5A, overexposed westerns –again replicates are required. The authors need to show that the low levels of *tim* mRNA at night in inoculated animals are due to higher levels of TIM at night (more RNA repression, negative feedback). Again the *per* results are not consistent with a negative feedback model.

Response: In this revision, we provided the low exposed pictures. All these western blotting results have three biological repeats as shown in the source data as requested.

In non-inoculated insects, at the end of day, the protein level of TIM is very low, which result in increasing level of *tim* mRNA. At the early night, the increasing level of *tim* mRNA translate into high level of TIM protein, which then inhibit *tim* mRNA at the late night. Thus, the *tim* mRNA level should increase at the early night and then decrease at late night. Similar results have been reported in other insect studies as following (Wang, et al., 2021; Grima et al., 2019).

In contrast, the inoculated SBPHs have lower levels of *tim* and *per* at night due to inhibited degradation of both TIM and PER by the P6–LsCSN5 interaction. Therefore, we conclude that virus infection inhibits TIM degradation and weakens the oscillation of *tim/per*.

References:

Wang G.D., et al. Clock genes and environmental cues coordinate Anopheles pheromone synthesis, swarming, and mating. *Science* 371, 411-415 (2021). doi: 10.1126/science.abd4359

Grima B, et al. PERIOD-controlled deadenylation of the timeless transcript in the *Drosophila* circadian clock. *Proc Natl Acad Sci U S A* 116, 5721-5726 (2019). doi: 10.1073/pnas.1814418116.

Fig 5b – 2-way ANOVA not multiple t-testing

Response: We analyses all the qPCR results using two-way ANOVA followed by Tukey's test. **Please see the new Fig.5.**

Fig 5c – lack of replicates. Absolute minimum of two required, three to generate an sem.

Response: Three independent repeats were shown in source data as requested by the journal. **Please see the source data.**

Fig 5f, 5g – quantification?

Response: Thanks. The fluorescence in the fig. 5g is difficult to be quantified. The RFP protein and BYSMV N protein could be quantified based on the gel intensities. **Please see new Fig. 5f. The fig 5g**

L248 dpi? days post infection?

Response: Correct. Dpi indicate days post infection/inoculation.

Fig 6 title ‘enhances’ rather than ‘improves’

Response: Corrected, thanks. **Please see new Fig. 6.**

Fig 6e ANOVA

Response: Changed as requested, thanks. Two-way ANOVA followed by Tukey’s test was performed to investigate the main effects of P6 and P6^{116A} expression on mRNA levels of *tim* and *per*. Differences were considered significant at $P < 0.05$. **Please see new Fig. 6.**

The locomotor data in Fig 6 also need to be collected in DD and LL to examine any effects on clock function. Simply observing elevated locomotor levels may not be anything to do with the clock itself – maybe just the locomotor output is affected.

Response: We agree with the reviewer and appreciate the helpful suggestion. We performed DAM2 assays of transgenic flies in DD and LL cycles. The DD rhythm was indeed affected by the transgenic P6 expression. In LL, as predicted by the reviewer, the flies expressing P6-GFP displayed a rhythmic locomotor pattern probably due to the P6-mediated compromised degradation of TIM. Whereas, other flies lost rhythmic

locomotor activity due to light-triggered TIM degradation. Please see the new supplemental Fig. S9 and S10.

Supp Fig 6a not necessary

Response: Deleted, thanks.

L280 The authors claim high levels of arrhythmicity in the overexpressed P6 flies but do not give their criteria for arrhythmicity. Was this simply by eyeballing the LD patterns? A spectral analysis of the data is required.

Response: The determination of arrhythmicity was obtained by FaasX analysis software (Chiu et al., 2010). Rhythmic flies were defined by χ^2 periodogram analysis with the following criteria: power ≥ 20 and width ≥ 1.5 . Please see Line 452-455 and new supplementary Fig.8.

Reference: Chiu, J.C., et al., (2010). Assaying locomotor activity to study circadian rhythms and sleep parameters in *Drosophila*. *Journal of visualized experiments: JoVE*, (43), 2157.

In conclusion, while the results are interesting, I do not see the connection between hyperactivity, increased feeding and clock dysfunction as proven. Perhaps the products of locomotor activity output genes are targeted directly by COP9 mediated degradation? Furthermore, many of the results need replication of westerns particularly with associated statistical analyses and locomotor experiments to be performed in constant conditions, DD and LL. The results also do not conform to predictions based on the negative feedback model of insect clocks. There is work by Sehgal's group showing that Clock mutants also show hyperactivity. If CLOCK is targeted by the virus, this could explain why both *per* and *tim* transcripts are affected equally in inoculations.

Response: Thanks for reviewer insightful summary. Previous studies have demonstrated that the feeding behavior and locomotor activity are regulated by the circadian clock in *Drosophila* (Allada and Chung, 2010; Xu et al., 2008). It is generally

assumed that the core PER/TIM oscillatory loop triggers a rhythmic expression of various genes for modulation of insect behavior and physiological functions.

Repeats of western blotting and locomotor experiments of DD and LL have been shown in source data as requested.

As suggested, we examined accumulation of the *clk* and *cyc* transcripts and found that virus infections also weaken the oscillation of *clk/cyc*. In our view, BYSMV may not directly target CLOCK, but affect the TIM protein degradation and indirectly weaken the oscillation of *clk/cyc*. The Sehgal group's studies have shown that the *clock* mutants display hyperactivity (Zheng et al., 2009). In consistence, our results reveal that BY-RFP infection or P6 transgene induce hyperactivity through weaken the oscillation of clock-related genes, and then enhance virus transmission. **Please see the proposed model in the new Fig. 7.**

References:

Allada, R., and Chung, B.Y. (2010). Circadian organization of behavior and physiology in *Drosophila*. *Annu. Rev. Physiol.* 72, 605–624.

Xu, K., et al., (2008). Regulation of feeding and metabolism by neuronal and peripheral clocks in *Drosophila*. *Cell metabolism*, 8, 289–300.

Zheng, X., et al. (2009). An isoform-specific mutant reveals a role of PDP1 epsilon in the circadian oscillator. *J Neurosci.* 29, 10920–10927.

Reviewer #2 (Remarks to the Author):

Here the authors use a RFP-tagged BYSMV and localize it in intact insects (Fig. S1) and in tissue sections (Fig. 1b). From Fig. S1a it seems that the RFP label is homogeneous all over the insect, whereas in Fig. 1b it is rather punctuate and there is a big RFP-labeled aggregate in the lower left corner, whereas the head is on the right side and contains less label. Fig 1c shows RFP label in the brain of dissected heads, but not

in salivary glands. Why is this so? Has the virus not yet invaded the salivary glands at 12 days? On the other hand, there is a strong label, apparently stronger than in the brain, in a structure next to the gland. Please discuss this.

Response: The different signal intensity is due to distinct methods. In the Fig. S1a, we took photos of BY-RFP-infected SBPHs using an Olympus stereomicroscope SEX16 (Olympus, Tokyo, Japan). Since BY-RFP mainly infects the muscle and hemolymph, leading to strong RFP signal from the whole bodies.

In contrast, in Fig. 1b, histological analysis need dehydration, embed, section, and then re-hydration processes, which attenuate the RFP signal significantly. In addition, confocal microscopy images just show one slide (8 μm) of longitudinal section, so the signal is weak than the whole insect. The punctuate signal is from infected nerve fibers.

In the lower left corner, the strong RFP signal is from partial of alimentary canal that accumulate high level of virus. Indeed, the strong RFP signal next to the brain is accessory salivary glands. By contrast, BYSMV infect principal salivary glands at a very low efficiency. All these results are consistent with our previous results (Cao et al., 2018). In this study, we just focused on the brain infection, which is related to locomotor activity.

Note: Internal organs of infected SBPHs were immunolabeled for BYSMV with N-FITC (green) and stained for actin with phalloidin-rhodamine (red) from Cao et al., 2018.

Refer: Cao Q, et al., (2018) Transmission characteristics of barley yellow striate mosaic virus in its planthopper vector *Laodelphax striatellus*. *Frontiers in Microbiology* 9: 1419.

Then locomotor activity is assessed by DAM2 and shows clearly an increase in activity of infected insects. I do not understand though, what the individual columns along the timeline (1 to 5 is for 1 to 5 days, I assume) in Fig. 1e present exactly (I assume it is some kind of movement activity, but I did not find the information neither in materials and methods, nor in the figure because there is no labeling (neither unit nor parameter) of the y-axis.

Response: We apologize for these confusions. We have enriched the description of the method and added the label of the y-axis. The 1 to 5 indicate 1 to 5 days for monitoring the locomotor activity of insects continuously. The column numbers represent total across times of indicated SBPHs ($n = 32$) in the middle of tubes monitored by infrared beam in 5 d of LD cycles. We added the detail information in the figure legends and method. In addition, we added histograms to indicate the distribution of activity through 24 h. Please see the new fig. 1 and its figure legend.

Then the feeding behavior is measured by EPG and honeydew assay and shows increased phloem feeding of infected insects, compatible with increased virus inoculation. However, it is unclear whether the planthoppers were placed on healthy or infected plants for EPG and honeydew analysis. Please add this relevant information.

Response: Thank you for your suggestion. Here, we used the healthy barley plants for EPG assays. We have added the information. Please see Lines 154 and 161.

RT-qPCR and western blotting analysis indicates that TIM and PER oscillation is impaired in infected insects. This could be due to interaction of BYSMV protein P6 with planthopper protein CSN5, which is then tested and verified by several interaction assays. Evidence for a functional role of this interaction is shown by the fact that P6 inhibits CSN5-dependent deneddylation of CUL1, a component of the proteasome degradation machinery. Deneddylation is required for degradation activity, therefore P6 could inhibit CUL1-dependent degradation-mediated oscillation of TIM and PER levels by interacting with CSN5. The authors present evidence for this by silencing CSN5 and measuring TIM and PER levels. Further they show that this correlates with

increased locomotion and transmission. Finally, P6 expression in *Drosophila* has similar effects on locomotion activity, TIM and PER accumulation as in planthoppers, further supporting the authors' hypothesis on P6 action.

Similar results of behavior changes of virus-infected planthoppers and other plant virus vectors have been reported quite often. Relating it with specific changes in the vector has only been reported rarely. This makes this paper really interesting and the authors present convincing evidence that viral P6 interacts with a component of the proteasome degradation pathway and inhibits degradation of circadian oscillation-relevant TIM/PER, thus provoking the behavior changes relevant for more efficient virus transmission.

Response: We thank the reviewer for insightful summary on our results.

What is less clear, is the microscopic data. Here the methods are, in my opinion, not sufficiently well described: microscopy settings and acquisition conditions (stacks or optical sections, filter settings etc.) are not mentioned; most acquisitions seem to show images of whole body or dissected organs, but in materials and methods there is also mentioning of tissue sections. I assume this applies to Fig. 1b, but you should indicate which images show sections. In Fig. 1c and S2, the label is presented in reddish-yellow. Why is this label in yellow, if it is confocal RFP? Are these confocal false color presentations, confocal superpositions of different acquisition channels or are these multicolor epifluorescence acquisitions? Please resolve this issue and describe more precisely the acquisition conditions.

Response: Thank you for your suggestion. We added the microscopy setting and acquisition conditions in the figure legends and method.

In Fig 1b, longitudinal sections of whole bodies was monitored in a z-stack model with an optical slice thickness of 3 \$\mu\text{m}\$ using a \$\times 10\$ 0.32 N.A. objective. The detailed localization in brains was acquired with an optical slice thickness of 0.3 \$\mu\text{m}\$ using a \$\times 100\$, 1.44 N.A. oil objective.

In Fig. 1c and S2, we used the BX53 microscope using the DP72 digital camera (Olympus, Tokyo, Japan), rather than a confocal microscope. Therefore, we took the photos using different acquisition channels (Bright or RFP) subsequently. The color indicates true color, rather than false color. Since virus accumulate to very high level in salivary gland, high intensity of RFP usually exhibits reddish-yellow under fluorescence microscope.

All the detail information of acquisition conditions was provided in the revised method. Please see Line 410-442.

Also, the claims of RFP localization seem not always be fully supported by the shown images (see above). The use of free RFP as a marker might result in misleading localization information of the virus, because it can freely diffuse. So while this approach allows clearly to distinguish between infected and healthy planthoppers, precise localization might be difficult to assess. The authors should consider this.

Response: Thank you for your suggestion. Here, we wish to point out that the RFP protein was only expressed in the BY-RFP infected cells. In addition, RFP has no secretion signal. Therefore, the RFP signal can be used as indicator of virus-infected cells or tissues. For examples, BY-RFP only infects accessory salivary glands, rather than the principal salivary glands in Fig. 1c, which can rule out free diffusion of RFP. Please see the Fig. 1.

We also dissected brain tissues and stained with the specific rabbit antibodies against the N protein and then incubated with alex-488-coupled rabbit secondary antibody (GFP) for observation under a fluorescence microscopy. As expected, RFP fluorescence from BY-RFP was finely overlapped with the viral N protein in the viruliferous SBPHs, rather than in those of non-inoculated controls. These results indicate that RFP fluorescence can used as indicator of virus infection sites. Please see the new Supplementary Fig. S3.

The authors should also describe more in detail the recombinant virus, in particular the RFP insertion; the information in Fig. 1a is not sufficient.

Response: Agree. We added the detail information as following “To this end, we used a recombinant BYSMV expressing red fluorescent protein (BY-RFP), in which the RFP gene with the gene junction sequences was inserted between the N and P genes of BYSMV (Fig. 1a)39. BY-RFP can express RFP protein during virus infection, which allows to visualize authentic infection of BYSMV in living tissues of SBPHs.” The detail rescue process has been described in our previous report (Gao, et al., 2019).

Reference:

Gao Q, et al., (2019) Rescue of a plant cytorhabdovirus as versatile expression platforms for planthopper and cereal genomic studies. *New Phytologist* 223: 2120-2133. <https://doi.org/10.1111/nph.15889>.

Note: The Figure 1 of Gao et al., *New Phytologist*, 2019

Concerning the discussion, there is some data that other plant viruses modify the nervous system of insect vectors (for example PMID:32729829), and that also other mechanisms might apply to induce transmission-friendly behaviors (for example PMID:35246603).

Response: As suggested, we cited the two excellent papers related with insect nervous modification and virus transmission in the introduction section. **Please see line 61-65.**

Reference:

Wang SF, et al., (2020) Apoptotic neurodegeneration in whitefly promotes the spread of TYLCV. *eLife* 9: e56168.

Wu W, et al., (2022) A leafhopper saliva protein mediates horizontal transmission of viral pathogens from insect vectors into rice phloem. *Commun. Biol.* 5: 204.

Another point you might want to address: since P6 interacts with CSN5 and indirectly with CUL1, could it also inhibit degradation of other targets than TIM/PER?

Response: We appreciate and agree with these comments. We added the following statement in the discussion: “Thus, we assume that P6–LsCSN5 interaction inhibit the CUL1-based E3 ligase-mediated degradation of other targets than the TIM protein, which would contribute to other important functions of BYSMV P6 in virus infections.”

Please see line 370-373.

Other comments

L 104 Wording a bit strange ‘alimentary canals’

Response: Alimentary canal is a long enclosed coiled tube through insect bodies that is divided into three regions, foregut, midgut, and hindgut, with a different function in digestion. The word has been widely used in insect science as the following reference.

Most plant viruses enter alimentary canals along with feeding and initially infect epidermal cells of alimentary canals.

Reference: Li S, Jing X. Fates of dietary sterols in the insect alimentary canal. *Curr. Opin. Insect Sci.* 2020, 41:106-111. doi:10.1016/j.cois.2020.08.001

L 105 Please indicate how the virus was RFP-tagged. From Fig. 1A it seems that it is free RFP, but please confirm. I did not find information on cloning in material and methods.

Response: Thank you for your suggestion. The *RFP* gene is inserted into the cDNA infectious clone of BYSMV. Thus, the RFP protein is only expressed along with the replication of BYSMV in host plants and insect vectors. The detail information has provided above.

L 128 It rather increases than induces activity.

Response: Replaced, thanks. Please see line 137.

L 276 'Files' should read 'flies'.

Response: Corrected, thanks. Please see line 300.

Fig. 1b: Bright field images might be useful to illustrate virus localization.

Response: As suggested, we obtained bright field images that are not informative, so we did not show the bright field in figures.

Fig. 6b: Fluorescent imaging of whole flies might be nice to show exclusive brain localization of P6.

Response: Thanks for your suggestion. However, it is difficult to observe the fluorescence from the heads of whole flies, because the *elav-Gal4*-driven P6-GFP was specifically expressed in the nervous systems. Many studies used dissected CNS for GFP observation as the following paper (Ogienko et al. 2020; Berger et al. 2007).

Reference:

Ogienko AA, et al., (2020) Molecular and cytological analysis of widely-used Gal4 driver lines for *Drosophila* neurobiology. *BMC Genet* 21: 96. <https://doi.org/10.1186/s12863-020-00895-7>

Berger C, et al., (2007) The commonly used marker ELAV is transiently expressed in

neuroblasts and glial cells in the *Drosophila* embryonic CNS. *Dev Dynam* 236: 3562-3568. <https://doi.org/10.1002/dvdy.21372>

Reviewers' Comments:

Reviewer #1:

Remarks to the Author:

The authors should be congratulated on producing a beautiful, clever and informative paper. They have incorporated most of my comments. I have a couple more suggestions that are mostly minor

L54-55 -implications of aggressive bees for virus transmission?

L140-144 no difference in LL activity – maybe but I think some stats are required- you cannot just eyeball data –for each 30 min bin are necessary for the reader to see the variability of the data and a 2-way ANOVA taking average activity per animals under DD and LL with and without virus should give a nice interaction, showing the differences the authors are eyeballing. Very simple to do – would take 10 mins.

L159. I think the authors mean Suppl Fig 5 not 3

Fig 3a-d That clk/cyc transcripts oscillate in antiphase to per/tim is to be expected under the negative feedback model. I do not understand why TIM cycles in phase with its transcript with a peak at ZT18? Perhaps some comment required.

L187 TIM protein is not shown in Fig 3c but in Fig 3e

Fig 5d Locomotor activity indicated by total activity counts (y-axis) every 1 h (n = 18).??? What does that mean? Every 1h??? The results are presented in 4 h time windows. Again ANOVA (2-way is required see line 264-267)

I have no other comments.

Reviewer #2:

Remarks to the Author:

The authors' replies and modifications of the work in response to my comments are fine with me. Only smaller items for more clarity:

Please add to the legend of figure 1 that 1b and 1c are confocal image stacks, to the legend of figure 5h "Representative brightfield and epifluorescence images of RFP fluorescence..."

Please add to the legend of figure S1 that it is epifluorescence microscopy, in the legend to figure S2 replace "observed with an olympus immunofluorescence" by "observed by brightfield and epifluorescence microscopy", replace in the legend of figure S3 "under fluorescence microscopy" by "by brightfield and epifluorescence microscopy".

REVIEWER COMMENTS

Reviewer #1 (Remarks to the Author):

The authors should be congratulated on producing a beautiful, clever and informative paper. They have incorporated most of my comments. I have a couple more suggestions that are mostly

minor.

Response: Many thanks for your positive comments.

L54-55 -implications of aggressive bees for virus transmission?

Response: In the cited paper, the authors show that a novel picorna-like virus only was detected in aggressive bee but not in nurse bees or foragers, indicating a close relation between viral infection in the brain and aggressive worker behaviors. We added the following statement “implying that honeybee aggressive behaviors probably enhance virus horizontal transmission”. Please see line 56-57.

Refer: Fujiyuki T, et al. Novel insect picorna-like virus identified in the brains of aggressive worker honeybees. Journal of Virology 78, 1093–1100 (2004).

L140-144 no difference in LL activity – maybe but I think some stats are required- you cannot just eyeball data –for each 30 min bin are necessary for the reader to see the variability of the data and a 2-way ANOVA taking average activity per animals under DD and LL with and without virus should give a nice interaction, showing the differences the authors are eyeballing. Very simple to do – would take 10 mins.

Response: Agree and thanks for your insightful suggestion. We reanalyzed the results the supplementary Fig. 4. Two-way ANOVA followed by Tukey’s test was performed to investigate main effects of virus infection on insect activity. Differences were considered significant at $P < 0.05$. The results indicate that constant light treatment abolished the effect of virus infection on insect locomotor activity. Please see line 145-146 and new Supplementary Fig. S4.

L159. I think the authors mean Suppl Fig 5 not 3

Response: Corrected, thanks! Please see line 161.

Fig 3a-d That *clk/cyc* transcripts oscillate in antiphase to *per/tim* is to be expected under the negative feedback model. I do not understand why TIM cycles in phase with its transcript with a peak at ZT18? Perhaps some comment required.

Response: According to the feedback loop of circadian clock, at the end of day (ZT 0), TIM is subjected to ubiquitination and proteasomal degradation through the CSN5-regulated CUL1-based E3 ligases. Thereafter, without inhibition of PER/TIM heterodimer, two transcription factors, CLOCK (CLK) and CYCLE (CYC), form a heterodimer and promote transcription of period (*per*) and timeless (*tim*) genes at ZT 12-18, leading to the increasing mRNA levels of *per* and *tim*. The new translated PER/TIM proteins in turn inhibit CLK/CYC-mediated transcription of *per* and *tim* genes, resulting in gradually decreased mRNA levels of *per* and *tim* after ZT 18. Therefore, TIM and its transcript exhibit a peak at ZT18. Similar results have been reported in other insect studies (PMID: 33479155; PMID: 30833404; PMID: 30059997).

L187 TIM protein is not shown in Fig 3c but in Fig 3e

Response: Corrected, thanks! Please see line 189-190.

Fig 5d Locomotor activity indicated by total activity counts (y-axis) every 1 h (n = 18).??? What does that mean? Every 1h??? The results are presented in 4 h time windows. Again ANOVA (2-way is required see line 264-267)

Response: Locomotor activity indicated by total activity counts (y-axis) of 32 insects in 6 h. Two-way ANOVA was performed to investigate effects of *dscsn5* on insect activity ($P < 0.0001$). Asterisks indicate significant differences between *dsgfp*- and *dscsn5*-treated at same time points. Please see line 863-867 of Fig 5 legend.

I have no other comments.

Response: we appreciate the reviewer's effort and insightful comments.

Reviewer #2 (Remarks to the Author):

The authors' replies and modifications of the work in response to my comments are fine with me.

Response: Many thanks for your positive comments.

Only smaller items for more clarity:

Please add to the legend of figure 1 that 1b and 1c are confocal image stacks, to the legend of figure 5h “Representative brightfield and epifluorescence images of RFP fluorescence...”

Response: Thanks! We have added the suggested information in the legends of Figure 1 and 5. Notably, the updated figure 1c is representative confocal image but not image stacks, which clearly show the presence of BY-RFP in brain tissue. Please see the updated figure legends of figure 1 and figure 5 in PAGE 30 and 35.

Please add to the legend of figure S1 that it is epifluorescence microscopy, in the legend to figure S2 replace “observed with an olympus immunofluorescence” by “observed by brightfield and epifluorescence microscopy”, replace in the legend of figure S3 “under fluorescence microscopy” by “by brightfield and epifluorescence microscopy”.

Response: Thanks! We have added the suggested information “observed with a brightfield and epifluorescence microscopy” in the legends of Figure S1, S2, and S3.